# Di$^2$Pose: Discrete Diffusion Model for Occluded 3D Human Pose Estimation

**Weiquan Wang**[1], **Jun Xiao**[1], **Chunping Wang**[2], **Wei Liu**[3], **Zhao Wang**[1], **Long Chen**[4]*

[1]Zhejiang University   [2]Finvolution Group   [3]Tencent
[4]Hong Kong University of Science and Technology

{wqwangcs, junx}@zju.edu.cn, wangchunping02@xinye.com,
wl2223@columbia.edu, zhao_wang@zju.edu.cn, longchen@ust.hk

## Abstract

Diffusion models have demonstrated their effectiveness in addressing the inherent uncertainty and indeterminacy in monocular 3D human pose estimation (HPE). Despite their strengths, the need for large search spaces and the corresponding demand for substantial training data make these models prone to generating biomechanically unrealistic poses. This challenge is particularly noticeable in occlusion scenarios, where the complexity of inferring 3D structures from 2D images intensifies. In response to these limitations, we introduce the **Di**screte **Di**ffusion **Pose** (**Di$^2$Pose**), a novel framework designed for occluded 3D HPE that capitalizes on the benefits of a discrete diffusion model. Specifically, Di$^2$Pose employs a two-stage process: it first converts 3D poses into a discrete representation through a *pose quantization step*, which is subsequently modeled in latent space through a *discrete diffusion process*. This methodological innovation restrictively confines the search space towards physically viable configurations and enhances the model's capability to comprehend how occlusions affect human pose within the latent space. Extensive evaluations conducted on various benchmarks (*e.g.*, Human3.6M, 3DPW, and 3DPW-Occ) have demonstrated its effectiveness.

## 1   Introduction

3D Human Pose Estimation (HPE) from monocular images remains a challenging yet pivotal research in the realm of computer vision, boasting a wide range of applications including human-machine interaction, autonomous driving, and animations [57, 81, 5, 70]. Generally, the mainstream approaches, including Direct Estimation [68, 43, 47] and 2D-to-3D Lifting [87, 51, 86], aim to perform 3D HPE by either directly predicting 3D poses from 2D images or lifting detected 2D poses into 3D space. These approaches aim to address the inherent 2D-3D ambiguity in 3D HPE tasks by learning mapping from training data. Despite significant advancements, accurately estimating 3D poses from monocular images remains a formidable challenge, particularly when humans are partially occluded [39]. Such occlusions introduce considerable uncertainty and indeterminacy into the estimation process.

Existing 3D HPE methods try to handle the occlusion challenges with pose priors/constraints [58, 62] or data augmentation strategies (*e.g.*, annotations augmentation [61], pose transformation [35], and differentiable operations [82]). However, due to the inherent discreteness of 3D poses (primarily defined by discrete anatomical landmarks), these methods tend to represent poses using coordinate vectors or heatmap embeddings, treating joints as independent units and overlooking the interdependencies among body joints. Recent research [21] has introduced a compositional pose representation that captures the dependencies among joints by converting a pose into multiple tokens, enabling the

---

*Long Chen is the corresponding author.

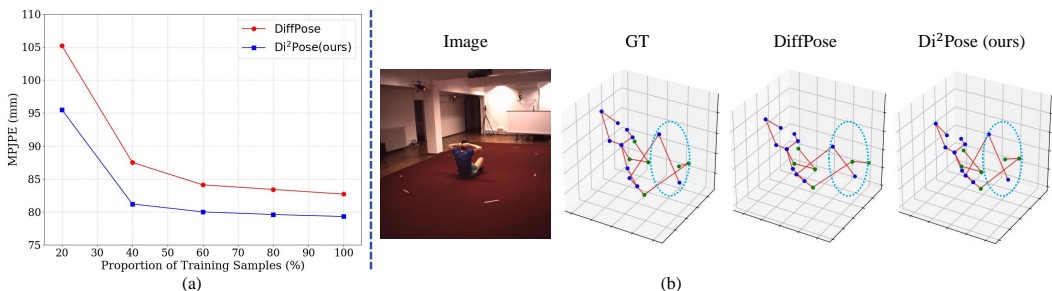

Figure 1: (a) Results of DiffPose [22] and Di$^2$Pose in Human3.6M [72] dataset (with MPJPE metric), across varying proportions of training samples. (b) Prediction results of two methods under occlusion.

use of mutual context between joints. This approach, which learns from real pose datasets, results in each learned token corresponding to a physically realistic prototype. Nevertheless, Geng *et al.* [21] casts HPE to a classification task, where the system simply classifies tokens based on prototype poses. Unfortunately, such scheme does not account for the effects of occlusions in the estimation process, potentially leading to inaccuracies due to unresolved uncertainty and indeterminacy.

Recent studies have shown marked progress in 3D HPE via generative models [2, 84, 19, 27]. Notably, diffusion models [25] have demonstrated effectiveness in handling complex and uncertain data distributions, making them suitable for handling uncertainty and indeterminacy in 3D HPE tasks [19, 91, 66, 27, 37]. They excel at generating samples that conform to a target data distribution by iteratively removing noise through a series of diffusion steps, ultimately predicting more accurate 3D poses. However, these diffusion-based 3D HPE methods initialize the 3D pose from random noise at the begining of the diffusion process, where each joint can be sampled from the continuous 3D space. Since the continuous 3D space has an infinite number of points, training such diffusion-based models requires a large amount of 3D pose data to achieve optimal outcomes [23, 75, 3]. This demand implies a substantial need for training data, presenting a stark contradiction to the limited availability of 3D human pose datasets. As illustrated in Figure 1(a), the predictive performance of DiffPose [22] declines more rapidly as the proportion of training data decreases. Given the scarcity of 3D pose training data, previous diffusion models may generate physically implausible configurations that do not adhere to human biomechanics, leading to inaccurate human pose estimations, particularly in occluded scenes (*c.f.*, Figure 1(b) with DiffPose).

In this paper, we propose a novel framework for 3D HPE with occlusions: **Di**screte **Di**ffusion **Pose** (**Di**$^2$**Pose**), drawing on compositional pose representation and diffusion model to achieve the best of two worlds. Specifically, Di$^2$Pose employs a two-stage approach: *a pose quantization step* followed by *a discrete diffusion process*. The pose quantization step leverages the discrete nature of 3D poses and represents them as quantized tokens by capturing the local interactions between joints. This step effectively confines the search space to physically plausible configurations by learning from real 3D human poses. Subsequently, the discrete diffusion process models the quantized pose tokens in the latent space through a conditional diffusion model. By integrating the forward and reverse processes, our framework adeptly simulates the transition of a 3D pose from occluded to recovered. By modeling occlusion implicitly within the latent space, Di$^2$Pose enhances its understanding of how occlusions affect human poses, providing valuable insights during the training phase.

**For the pose quantization step**, we devise a pose quantization step inspired by VQ-VAE [71], consisting of a pose encoder, a quantization process, and a pose decoder. To effectively capture local interactions between 3D joints, we introduce the Local-MLP block for both pose encoder and decoder. Within each Local-MLP block, a simple Joint Shift operation integrates information from different joints. The pose encoder utilizes several Local-MLP blocks to convert a 3D pose into multiple rich token features, each representing a sub-structure of the overall pose. These tokens are quantized using a shared codebook, yielding corresponding discrete indices. Additionally, we implement the finite scalar quantization (FSQ) [49] to address the codebook collapse issue observed in traditional VQ-VAE methods [59, 89, 56, 42]. This strategy ensures that the generated codewords are meaningful, a crucial aspect for the subsequent success of the discrete diffusion process.

**For the discrete diffusion process**, during the training phase, we introduce `occlude` and `replace` strategies to model the quantized pose tokens, enabling the discrete diffusion model to predict occluded tokens and update potential tokens. The occluded token represents the occlusion of the corresponding sub-structure of the 3D pose. The token replacement mechanism is designed to enhance the diversity of potential sub-structures, reflecting the indeterminacy in occluded parts. During the inference phase, pose tokens are either occluded or initialized randomly. The denoising diffusion process estimates the probability density of pose tokens step-by-step based on the input 2D image until the tokens are completely reconstructed. Each step leverages contextual information from all tokens of the entire pose as predicted in the previous step, facilitating the estimation of a new probability density distribution and the prediction of the current step's tokens. This sequential approach ensures a detailed and accurate reconstruction of 3D poses from occluded scenes.

We extensively evaluate our approach in 3D HPE on three challenging benchmarks (Human3.6M [34], 3DPW [72] and 3DPW-Occ). Di$^2$Pose consistently yields lower errors compared to state-of-the-art methods. In particular, it achieves significantly better results when evaluated on occluded scenarios, verifying its advantages of occlusion-handling capability. Our contributions are threefold:

- We propose the Di$^2$Pose framework, which integrates the inherent discreteness of 3D pose data into the diffusion model, offering a new paradigm for addressing 3D HPE under occlusions.

- The designed pose quantization step represents 3D poses in a compositional manner, effectively capturing local correlations between joints and confining search space to reasonable configurations.

- The constructed discrete diffusion process simulates the complete process of a 3D pose transitioning from occluded to recovered, which introduces the impact of occlusions into pose estimation process.

## 2   Related Work

**Monocular 3D HPE.** Existing approaches can generally be classified into frame-based and video-based methodologies. **Frame-based methods** predict the 3D pose from a single RGB image, employing different networks in various studies [18, 20, 52, 54, 55] to directly output the human pose from the 2D image. Alternatively, a significant number of studies [48, 77, 85, 88] initially determine the 2D pose, which subsequently forms the foundation for inferring the 3D pose. In contrast, **video-based methods** leverage temporal relationships across video frames. Such methods predominantly [9, 11, 15, 31, 65, 73, 76] commence with the extraction of 2D pose sequences using a 2D pose detector from the video clips, aiming to harness the essential spatio-temporal data for 3D pose estimation. To validate the efficacy of our approach, we evaluate our Di$^2$Pose on the more challenging frame-based setting, wherein the 3D human pose is directly inferred from a 2D image.

**Occluded 3D HPE.** Occlusions significantly challenge 3D HPE. As evidenced by research [76, 58, 62], pose priors and constraints have been proven crucial for mitigating such issue. Approaches typically involve statistical models to deduce occluded parts from visible cues [76, 41, 44] or pre-defined rules to constraint poses [61, 1]. Moreover, due to the scarcity of 3D pose data, data augmentation, including synthetic occlusions [7, 38, 60, 64, 13] and pose transformations [35, 82], remains vital for enhancing model robustness. Diverging from these aforementioned methods, our method innovatively introduces occlusion in the latent space without extra priors or explicit augmentations, providing a deeper feature-based understanding of occlusion's effects on pose estimation.

**Diffusion Models for 3D HPE.** Recent advancements have shown that diffusion models are capable of managing complex and uncertain data distributions [25, 26, 17, 4, 32, 8, 10, 80, 40, 36, 74, 78], which is particularly beneficial for 3D HPE. Typically, these models predict 3D poses by progressively refining the pose distribution from high to low uncertainty [22, 14, 19, 91, 63]. Other approaches use diffusion models to generate multiple pose hypotheses from a single 2D observation [66, 27]. These 3D pose estimators effectively reduce uncertainty and indeterminacy throughout the estimation process. Moreover, discrete diffusion models have also gained attention in various domains [30, 40, 24, 33]. Inspired by these advancements, our work introduces a discrete diffusion model for occluded 3D HPE, which aligns more closely with the inherent discreteness of 3D pose data and effectively incorporates occlusion into the estimation process, providing a novel perspective in the field.

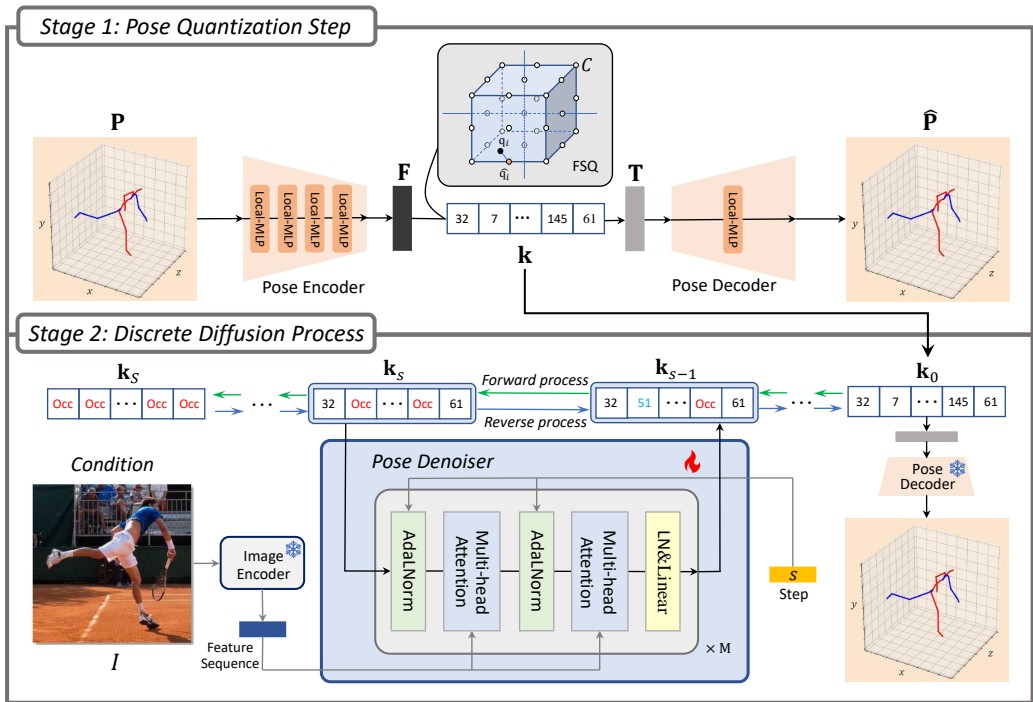

Figure 2: Overview of our two-stage Di²Pose framework. In the stage 1, we train a pose quantization step that transforms a 3D pose $\mathbf{P}$ into multiple discrete tokens $\mathbf{k}$, each token representing the indices of implied codebook $\mathcal{C}$. In the stage 2, we model $\mathbf{k}$ in the discrete space by discrete diffusion process. In the forward process, each token is probabilistically occluded with Occ token or replaced with another available token. In the reverse process, the model leverages an independent image encoder and a pose denoiser to reconstruct all the tokens based on the condition 2D image. These reconstructed tokens are finally decoded by the pose decoder, resulting in the recovered 3D pose. Notably, we only update the parameters of pose denoiser, pose decoder and image encoder are frozen.

## 3 Di²Pose

Given an 2D image $I \in \mathbb{R}^{H \times W \times 3}$, the goal of 3D HPE is to predict $\hat{\mathbf{P}} \in \mathbb{R}^{J \times 3}$, which represents the 3D coordinates of all the $J$ joints of the human body. In this paper, we construct occluded 3D HPE task as a two-stage framework including the pose quantization step and the discrete diffusion process.

As shown in Figure 2, **in the training phase**, *Stage 1* learns a pose quantization step by a VQ-VAE like structure (Sec. 3.1), which is able to encode a 3D pose into multiple quantized tokens. *Stage 2* models quantized pose tokens in the latent space by the forward and reverse process of a conditional diffusion model (Sec. 3.2). **In the inference phase**, we only use *the reverse process* of Stage 2 and the pre-trained pose decoder of Stage 1 to recover 3D pose from the 2D image. Notably, pose tokens are either occluded or initialized randomly at the beginning of the inference phase. The model reconstructs all the tokens based on the condition 2D image step-by-step. These reconstructed tokens are finally decoded by the pre-trained pose decoder, resulting in the recovered 3D pose.

### 3.1 Pose Quantization Step

As depicted in Figure 2, a pose quantization step comprises a pose encoder, the quantization process, and a pose decoder. Initially, for a real 3D pose $\mathbf{P} \in \mathbb{R}^{J \times 3}$, the pose encoder $f_{PE}(\cdot)$ converts $\mathbf{P}$ to token features $\mathbf{F}$. During the quantization process, we utilize FSQ to quantize $\mathbf{F} = (\mathbf{f}_1, \mathbf{f}_2, \cdots, \mathbf{f}_N)$ $(\mathbf{f}_i \in \mathbb{R}^D)$ into tokens $\mathbf{T} = (\mathbf{t}_1, \mathbf{t}_2, \cdots, \mathbf{t}_N)$ $(\mathbf{t}_i \in \mathbb{R}^D)$. Finally, the quantized tokens $\mathbf{T}$ are decoded by the pose decoder $f_{PD}(\cdot)$ to reconstruct 3D pose $\hat{\mathbf{P}}$.

**Pose Encoder.** Considering the interdependencies among human body joints, our goal is to represent 3D poses in a compositional manner, moving away from reliance on coordinates vectors or heatmap

embeddings. The VQ-VAE architecture, incorporating MLP-Mixer blocks [69] within its encoder and decoder, has been proven effective in decomposing a pose into multiple token features, each corresponding to a sub-structure of the pose [21]. However, the MLP-Mixer block is designed to extract global information across all joints, which can not adequately capture the local relationships between joints within individual sub-structure.

In response to aforementioned limitation, we design Local-MLP block to capture the local interactions between 3D joints. The pose encoder $f_{PE}(\cdot)$, comprising several Local-MLP blocks, converts $\mathbf{P}$ to $N$ token features:

$$\mathbf{F} = (\mathbf{f}_1, \mathbf{f}_2, \cdots, \mathbf{f}_N) = f_{PE}(f_{emb}(\mathbf{P})), \tag{1}$$

where $f_{emb}(\cdot)$ embeds $\mathbf{P}$ to $\mathbf{P_{emb}} \in \mathbb{R}^{J \times D}$ by a linear layer.

As shown in Figure 3(a), a Local-MLP block is composed of a Layer Normalization layer, a Joint Shift block (JS-Block), a Channel MLP, and a residual connection. The JS-Block is specifically designed to capture local interactions among $X$ joints. It extracts features by linear projection, and the Joint Shift operation enables feature translation along joint connection directions. As shown in Figure 3(b), with the input $\mathbf{P}^{\top}_{\mathbf{emb}} \in \mathbb{R}^{D \times J}$, the feature is evenly divided into $X$ segments ($X = 3$ in the example), each segment being shifted incrementally by units from $-\lfloor X/2 \rfloor$ to $\lfloor X/2 \rfloor$. The central segment remains stationary, while the segments to the left and right are symmetrically shifted away from the center by up to $\pm\lfloor X/2 \rfloor$ units. Zero padding is used to maintain dimensionality. Features highlighted within the dashed box are selected for further linear projection. Finally, the Channel MLP processes these features channel-wise to facilitate information integration.

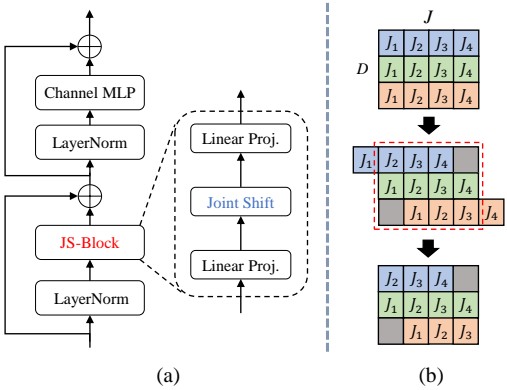

Figure 3: (a) depicts the structure of the Local-MLP block; (b) shows the Joint Shift operation, where the arrows indicate the steps, and different subscript numbers represent the features of different joints. The gray blocks indicate zero padding.

**Quantization Process.** During this process, we exploit FSQ [49] to enhance the utilization of codewords. FSQ quantizes token features $\mathbf{F}$ as corresponding token indices:

$$\mathbf{k} = (k_1, k_2, \cdots, k_N) = \text{FSQ}(f_{proj}(\mathbf{F})), \tag{2}$$

where $f_{proj}(\cdot)$ projects each $\mathbf{f}_i \in \mathbb{R}^D$ of $\mathbf{F}$ to $\mathbf{q}_i \in \mathbb{R}^d$, and each $k_i$ of $\mathbf{k}$ denotes the entries of implied codebook $\mathcal{C}$. For each $\mathbf{q}_i$, FSQ employs a bounding function $f_{bnd} : \mathbf{q}_i \mapsto \lfloor L/2 \rfloor \cdot \tanh(\mathbf{q}_i)$ to constrain each channel of $d$. As a result, each channel in $\hat{\mathbf{q}}_i = \text{round}(f_{bnd}(\mathbf{q}_i))$ takes one of $L$ unique values. This procedure yields $\hat{\mathbf{q}}_i \in \mathcal{C}$, where the total number of unique codebook entries is $|\mathcal{C}| = \prod_{i=1}^{d} L_i$ (mapping the $i$-th channel to $L_i$ values). The vectors in $\mathcal{C}$ can be enumerated, establishing a bijective mapping from any $\hat{\mathbf{q}}_i$ to an integer within $\{1, \ldots, |\mathcal{C}|\}$. In addition, the corresponding codeword of $k_i$, which is denoted as $\mathbf{t}_i \in \mathbb{R}^D$, represents the quantized result of $\mathbf{f}_i$. Thereby, using FSQ, the token features $\mathbf{F}$ are quantized as $\mathbf{T} = (\mathbf{t}_1, \mathbf{t}_2, \cdots, \mathbf{t}_N)$.

**Pose Decoder.** The pose decoder $f_{PD}(\cdot)$ is designed to recover 3D pose $\hat{\mathbf{P}}$ from $\mathbf{T}$. $f_{PD}(\cdot)$ adopts a structure similar to the pose encoder but in reverse, utilizing a reduced number of Local-MLP blocks.

**Loss.** The pose quantization step, including the pose encoder, quantization process, and pose decoder, is jointly optimized by minimizing L1 loss $\mathcal{L}_{PQ} = ||P - \hat{P}||_1$ across the training dataset.

### 3.2 Discrete Diffusion Process

After training the pose quantization step, we can acquire $N$ quantized tokens $\mathbf{k}$ from the original 3D pose $\mathbf{P}$. The next step in Di$^2$Pose pipeline is to model $\mathbf{k}$ in the latent space by the discrete diffusion process. In the following, we first briefly introduce the diffusion models and clarify the basic principles of the discrete diffusion model. Then we explain the details of discrete diffusion

process, including the designed transition matrix and loss function. Eventually, we illustrate the architecture and training and inference process.

**Discrete Diffusion Model.** Our discrete diffusion model is characterized by two distinct processes: 1) **Forward process**: It progresses through discrete steps $s \in \{0, 1, 2, ..., S\}$, gradually transforming the initial tokens $\mathbf{k}_0$ (the quantized token $\mathbf{k}$) into a noise-infused latent representation $\mathbf{k}_S$. 2) **Reverse process**: It is tasked with reconstructing the original data $\mathbf{k}_0$ from the latent $\mathbf{k}_S$, following a reverse temporal sequence $s \in \{S, S-1, ..., 1, 0\}$.

Followed previous studies [67, 3, 28], we use a transition probability matrix $[\mathbf{M}_s]_{ij} = q(\mathbf{k}_s = i | \mathbf{k}_{s-1} = j) \in \mathbb{R}^{|\mathcal{C}| \times |\mathcal{C}|}$ elucidate the likelihood of transitioning from $\mathbf{k}_{s-1}$ to $\mathbf{k}_s$. Then the forward process for the entire sequence of tokens is expressed as:

$$q(\mathbf{k}_s|\mathbf{k}_{s-1}) = \boldsymbol{c}^\top(\mathbf{k}_s)\mathbf{M}_s\boldsymbol{c}(\mathbf{k}_{s-1}), \tag{3}$$

where $\boldsymbol{c}(\cdot)$ symbolizes a function capable of converting a scalar into a one-hot column vector. The distribution of $\mathbf{k}_s$ follows a categorical distribution, determined by the vector $\mathbf{M}_s\boldsymbol{c}(\mathbf{k}_{s-1})$. Leveraging the Markov chain property, it is feasible to bypass intermediate stages, directly computing the probability of $\mathbf{k}_s$ from $\mathbf{k}_0$ for any given step as:

$$q(\mathbf{k}_s|\mathbf{k}_0) = \boldsymbol{c}^\top(\mathbf{k}_s)\overline{\mathbf{M}}_s\boldsymbol{c}(\mathbf{k}_0), \text{ with } \overline{\mathbf{M}}_s = \mathbf{M}_s \dots \mathbf{M}_1 \tag{4}$$

Moreover, the posterior of the reverse process, $q(\mathbf{k}_{s-1}|\mathbf{k}_s, \mathbf{k}_0)$, can be ascertained as:

$$q(\mathbf{k}_{s-1}|\mathbf{k}_s, \mathbf{k}_0) = \frac{q(\mathbf{k}_s|\mathbf{k}_{s-1}, \mathbf{k}_0)q(\mathbf{k}_{s-1}|\mathbf{k}_0)}{q(\mathbf{k}_s|\mathbf{k}_0)} = \frac{\left(\boldsymbol{c}^\top(\mathbf{k}_s)\mathbf{M}_s\boldsymbol{c}(\mathbf{k}_{s-1})\right)\left(\boldsymbol{c}^\top(\mathbf{k}_{s-1})\overline{\mathbf{M}}_{s-1}\boldsymbol{c}(\mathbf{k}_0)\right)}{\boldsymbol{c}^\top(\mathbf{k}_s)\overline{\mathbf{M}}_s\boldsymbol{c}(\mathbf{k}_0)}. \tag{5}$$

**Occlude and Replace Transition Matrix.** Notably, a suitable design for transition probability matrix $\mathbf{M}_s$ is significant to train the discrete diffusion process. As illustrated in Section 3.1, through pre-trained pose encoder and FSQ, $\mathbf{P}$ can be converted to $\mathbf{k} = (k_1, k_2, \cdots, k_N)$, each $k_i$ corresponding to a sub-structure of the overall pose. With this foundation, we specifically devise the `occlude` and `replace` scheme, which is inspired by [24], for tackling the challenges of occluded 3D HPE. In occlusion scenes, the human body is always occluded in various situations (self-occlusions, object or people-to-person occlusions), and the typical manifestation is that some sub-structures of the pose are invisible. Consequently, we design the `occlude` scheme simulating the occlusion of corresponding joints, which introduces occlusion impact in the training process. Additionally, recognizing the inherent uncertainty in occlusion scenarios where a single occluded region may correspond to multiple potential 3D human poses, we develop the `replace` strategy to update certain token with another available token.

In practice, each quantized token $k_i$ has a probability of $\gamma_s$ to transition to the `Occ` token. Moreover, $k_i$ is also subject to a probability of $|\mathcal{C}|\beta_s$ to be uniformly resampled across all $|\mathcal{C}|$ categories. Furthermore, $k_i$ retains a probability of $\alpha_s = 1 - |\mathcal{C}|\beta_s - \gamma_s$ to remain unchanged. Then, the transition matrix $\mathbf{M}_s \in \mathbb{R}^{(|\mathcal{C}|+1) \times (|\mathcal{C}|+1)}$ is defined as:

$$\mathbf{M}_s = \begin{bmatrix} \alpha_s + \beta_s & \beta_s & \cdots & 0 \\ \beta_s & \alpha_s + \beta_s & \cdots & 0 \\ \vdots & \vdots & \ddots & \vdots \\ \gamma_s & \gamma_s & \cdots & 1 \end{bmatrix}, \tag{6}$$

where $\alpha_s, \beta_s \in [0, 1]$. The prior distribution of step $S$ can be derived as: $p(\mathbf{k}_S) = [\overline{\beta}_S, \overline{\beta}_S, \cdots, \overline{\gamma}_S]$, where $\overline{\alpha}_S = \prod_{i=1}^S \alpha_i$, $\overline{\gamma}_S = 1 - \prod_{i=1}^S(1 - \gamma_i)$ and $\overline{\beta}_S = (1 - \overline{\alpha}_S - \overline{\gamma}_S)/|\mathcal{C}|$. In this study, we adapt the linear schedule [25] as noise schedule to pre-define the value of transition matrices ($\overline{\alpha}_S$, $\overline{\beta}_S$, and $\overline{\gamma}_S$). Subsequently, we can calculate $q(\mathbf{k}_s|\mathbf{k}_0)$ according to Eq. (4). However, when the number of categories $|\mathcal{C}|$ and time step $S$ is too large, it can quickly become impractical to store all of the transition matrices $\mathbf{M}_s$ in memory, as the memory usage grows like $O(|\mathcal{C}|^2 S)$. Actually, it is unnecessary to store all of the transition matrices. Instead we only store all of $\overline{\alpha}_s$ and $\overline{\beta}_s$ in advance, since we can calculate $q(\mathbf{k}_s|\mathbf{k}_0)$ according to following formula (refer to Appendix for proofs):

$$\overline{\mathbf{M}}_s\boldsymbol{c}(\mathbf{k}_0) = \overline{\alpha}_s\boldsymbol{c}(\mathbf{k}_0) + (\overline{\gamma}_s - \overline{\beta}_s)\boldsymbol{c}(|\mathcal{C}| + 1) + \overline{\beta}_s. \tag{7}$$

**Training Objectives.** We train a network $f_\theta(\mathbf{k}_{s-1}|\mathbf{k}_s, \boldsymbol{y})$ to estimate $q(\mathbf{k}_{s-1}|\mathbf{k}_s, \mathbf{k}_0)$ in the reverse process. The network is trained to minimize the variational lower bound (VLB):

$$\mathcal{L}_{vlb} = D_{KL}(q(\mathbf{k}_S|\mathbf{k}_0)||p(\mathbf{k}_S)) + \sum_{s=1}^{S-1}\left\{D_{KL}[q(\mathbf{k}_{s-1}|\mathbf{k}_s, \mathbf{k}_0)||f_\theta(\mathbf{k}_{s-1}|\mathbf{k}_s, \boldsymbol{y})]\right\}, \tag{8}$$

In addition, we follow [50, 24] to utilize the reparameterization trick, which lets Di$^2$Pose predict the noiseless token distribution $f_\theta(\hat{\mathbf{k}}_0|\mathbf{k}_s, \boldsymbol{y})$ at each reverse step, and then compute $f_\theta(\mathbf{k}_{s-1}|\mathbf{k}_s, \boldsymbol{y})$ as:

$$f_\theta(\mathbf{k}_{s-1}|\mathbf{k}_s, \boldsymbol{y}) = \sum_{\hat{\mathbf{k}}_0=1}^{H} q(\mathbf{k}_{s-1}|\mathbf{k}_s, \hat{\mathbf{k}}_0) f_\theta(\hat{\mathbf{k}}_0|\mathbf{k}_s, \boldsymbol{y}). \tag{9}$$

Based on the Eq. (9), an auxiliary denoising objective loss is introduced, which encourages the network to predict $f_\theta(\hat{\mathbf{k}}_0|\mathbf{k}_s, \boldsymbol{y})$:

$$\mathcal{L}_{k_0} = -\log f_\theta(\hat{\mathbf{k}}_0|\mathbf{k}_s, \boldsymbol{y}). \tag{10}$$

Our final loss function is defined as:

$$\mathcal{L} = \lambda\mathcal{L}_{\mathbf{k}_0} + \mathcal{L}_{vlb}, \tag{11}$$

where $\lambda$ is a hyper-parameter to control the weight of the auxiliary loss $\mathcal{L}_{\mathbf{k}_0}$.

**Diffusion Architecture.** As depicted in Figure 2, our discrete diffusion model consists of three main components: an image encoder, a pose denoiser, and a pose decoder. The pre-trained image encoder processes the 2D image to produce a conditional feature sequence. The pose denoiser, receiving the quantized pose tokens $\mathbf{k}_s$ and and step $S$, predicts the distribution of noiseless tokens $f_\theta(\hat{\mathbf{k}}_0|\mathbf{k}_s, \boldsymbol{y})$. This component is equipped with several transformer blocks, each featuring an AdaLNorm operator [6], multi-head attention blocks that combine the image feature information with $\mathbf{k}_s$, and layer normalization and linear layers. At the end of the reverse process, all recovered tokens are obtained, and the final prediction of 3D pose is decoded by the well-trained pose decoder.

**Training and Inference Process.** *In the training process*, as for step $s$, we sample $\mathbf{k}_s$ from $q(\mathbf{k}_s|\mathbf{k}_0)$ based on Eq. (7) in the forward process. We then estimate $f_\theta(\mathbf{k}_{s-1}|\mathbf{k}_s, \boldsymbol{y})$ in the reverse process. The final loss will be calculated according to Eq. (11). *In the inference process*, all pose tokens are either masked or initialized randomly. Subsequently, we predict $f_\theta(\mathbf{k}_{s-1}|\mathbf{k}_s, \boldsymbol{y})$ step by step until the tokens are completely recovered. Finally, reconstructed tokens are decoded by the pose decoder, resulting in the recovered 3D pose. The complete algorithms are summarized in Appendix.

## 4    Experiments

### 4.1    Datasets and Evaluation Metrics

**Datasets. Human3.6M** [34] is the most extensive benchmark for 3D HPE, consisting of 3.6 million images. We follow [22] with same protocol, which involves training on subjects S1, S5, S6, S7, and S8, and testing on subjects S9 and S11. **3DPW** [72] is the first dataset in the wild that includes video footage taken from a moving phone camera. We also evaluate our method on this dataset to measure the robustness and generalization. Additionally, to further verify the occlusion-robustness, we evaluate Di$^2$Pose on the **3DPW-Occ** [83], which is a subset of the 3DPW.

**Evaluation Metrics.** For Human3.6M and 3DPW, we follow the standard protocols. Mean per joint position error (**MPJPE**) calculates the mean Euclidean distance between the root-aligned reconstructed poses and ground truth joint coordinates. **PA-MPJPE** employs a Procrustes alignment between the poses before calculating the MPJPE. In addition, to further evaluate the effectiveness of our method under occlusion scenes, we devise an adversarial protocol, termed **3DPW-AdvOcc**, following the previous research [84]. We apply occlusion patches to the input image to identify the most challenging predictions. This process involves assessing the relative performance degradation on the visible joints. Similar to [84], we utilize textured patches generated by randomly cropping texture maps from the DTD [16]. We employ two square patch sizes: 40 and 80 relative to a 256 × 192 image, denoted as Occ@40 and Occ@80 respectively, with a stride of 10.

### 4.2    Implementation Details

**Pose Quantization Step.** The pose encoder is constructed with four Local-MLP blocks, while the pose decoder incorporates a single block. Within these Local-MLP blocks, the embedding dimensions $D$ for the pose encoder and decoder are configured to 2048 and 512, respectively. For the quantization process, the projected vector $\mathbf{q}_i$ features the channel $d = 5$. The levels per channel, denoted as $[L_1, \cdots, L_d]$, are specified as $[7, 5, 5, 5, 5]$. The number of quantized tokens $N$ is set to 100.

Table 1: Results on Human3.6M in millimeters under MPJPE. The best results are in **bold**, and the second-best ones are underlined.

| Methods | Dir | Disc | Eat | Gr. | Phon. | Phot. | Pose | Pur. | Sit | SitD. | Sm. | Wait | W.D. | Walk | W.T. | Avg |
|---|---|---|---|---|---|---|---|---|---|---|---|---|---|---|---|---|
| Pavlakos *et al.* [54] *CVPR'17* | 67.4 | 71.9 | 66.7 | 69.1 | 72.0 | 77.0 | 65.0 | 68.3 | 83.7 | 96.5 | 71.7 | 65.8 | 74.9 | 59.1 | 63.2 | 71.9 |
| Martinez *et al.* [48] *ICCV'17* | 51.8 | 56.2 | 58.1 | 59.0 | 69.5 | 78.4 | 55.2 | 58.1 | 74.0 | 94.6 | 62.3 | 59.1 | 65.1 | 49.5 | 52.4 | 62.9 |
| Hossain *et al.* [29] *ECCV'18* | 48.4 | 50.7 | 57.2 | 55.2 | 63.1 | 72.6 | 53.0 | 51.7 | 66.1 | 80.9 | 59.0 | 57.3 | 62.4 | 46.6 | 49.6 | 58.3 |
| Zhao *et al.* [85] *CVPR'19* | 48.2 | 60.8 | 51.8 | 64.0 | 64.6 | **53.6** | 51.1 | 67.4 | 88.7 | **57.7** | 73.2 | 65.6 | 48.9 | 64.8 | 51.9 | 60.8 |
| Liu *et al.* [45] *ECCV'18* | 46.3 | 52.2 | 47.3 | 50.7 | 55.5 | 67.1 | 49.2 | 46.0 | 60.4 | 71.1 | 51.5 | 50.1 | 54.5 | 40.3 | 43.7 | 52.4 |
| Xu *et al.* [77] *CVPR'21* | 45.2 | 49.9 | 47.5 | 50.9 | 54.9 | 66.1 | 48.5 | 46.3 | 59.7 | 71.5 | 51.4 | 48.6 | 53.9 | 39.9 | 44.1 | 51.9 |
| Zhao *et al.* [88] *CVPR'22* | 45.2 | 50.8 | 48.0 | 50.0 | 54.9 | 65.0 | 48.2 | 47.1 | 60.2 | 70.0 | 51.6 | 48.7 | 54.1 | 39.7 | 43.1 | 51.8 |
| Geng *et al.* [21] *CVPR'23* | — | — | — | — | — | — | — | — | — | — | — | — | — | — | — | 50.8 |
| Choi *et al.* [14] *IROS'23* | 44.3 | 51.6 | 46.3 | 51.1 | **50.3** | 54.3 | 49.4 | 45.9 | 57.7 | 71.6 | **48.6** | 49.1 | 52.1 | 44.0 | 44.4 | 50.7 |
| Zhang *et al.* [82] *TPAMI'23* | — | — | — | — | — | — | — | — | — | — | — | — | — | — | — | 50.2 |
| Gong *et al.* [22] *CVPR'23* | 42.8 | 49.1 | 45.2 | **48.7** | 52.1 | 63.5 | 46.3 | **45.2** | 58.6 | 66.3 | 50.4 | 47.6 | 52.0 | 37.6 | **40.2** | 49.7 |
| **Di²Pose (Ours)** | **41.9** | **47.8** | **45.0** | 49.0 | 51.5 | 62.2 | 45.7 | 45.6 | 57.6 | 67.1 | 50.1 | **45.3** | 51.4 | 37.3 | 40.9 | **49.2** |

Table 2: Evaluation on 3DPW, 3DPW-Occ, and 3DPW-AdvOcc. The number 40 and 80 after 3DPW-AdvOcc denote the occluder size. * denotes the results from our implementation. The best results are in **bold**, and the second-best ones are underlined.

| Methods | 3DPW [72] | | 3DPW-Occ [83] | | 3DPW-AdvOcc@40 | | 3DPW-AdvOcc@80 | |
|---|---|---|---|---|---|---|---|---|
| | MPJPE ↓ | PA-MPJPE ↓ | MPJPE ↓ | PA-MPJPE ↓ | MPJPE ↓ | PA-MPJPE ↓ | MPJPE ↓ | PA-MPJPE ↓ |
| Cai *et al.* [9] *ICCV'19* | 112.9 | 69.6 | 115.8 | 72.3 | 241.1 | 101.4 | 355.9 | 116.3 |
| Pavllo *et al.* [55] *CVPR'19* | 101.8 | 63.0 | 106.7 | 67.1 | 221.6 | 99.4 | 334.3 | 112.9 |
| Cheng *et al.* [12] *AAAI'21* | — | 64.2 | — | 85.7 | 279.4 | 113.2 | 371.4 | 119.8 |
| Zheng *et al.* [90] *ICCV'21* | 118.2 | 73.1 | 132.8 | 80.5 | 247.9 | 106.2 | 359.6 | 115.5 |
| Zhang *et al.* [82] *TPAMI'23* | 91.1 | 54.3 | 94.6 | 56.7 | 142.5 | 73.8 | 251.8 | 103.9 |
| Geng *et al.* * [21] *CVPR'23* | 83.1 | 53.9 | 82.8 | 53.7 | 127.2 | 71.9 | 192.5 | 92.1 |
| Gong *et al.* * [22] *CVPR'23* | 82.7 | 53.8 | 82.1 | 53.5 | 121.4 | 70.9 | 189.3 | 92.4 |
| **Di²Pose (Ours)** | **79.3** | **50.1** | **79.6** | **50.7** | **108.4** | **59.8** | **153.6** | **78.7** |

**Discrete Diffusion Process.** For the occlude and replace transition matrix, we linearly increase $\overline{\beta}_s$ and $\overline{\gamma}_s$ from 0 to 0.1 and 0.9, respectively, and decrease $\overline{\alpha}_s$ from 1 to 0. For the discrete diffusion model, we use off-the-shelf image encoder [79] to extract feature sequence of conditional 2D image. As for the pose denoiser, we build a 21-layer 16-head transformer with the dimension of 1024. We set steps $S$ as 100 and loss weight $\lambda$ is set to 5e-4. Please refer to Appendix for more details.

### 4.3 Comparsion with State-of-the-Arts

**Human3.6M.** To explore the effectiveness of Di²Pose, we evaluate its performance in the challenging context of frame-based 3D pose estimation. Specifically, within the discrete diffusion process, context information is extracted from a single input frame using the image encoder. As shown in Table 1, we benchmark Di²Pose against SOTA 3D HPE methods on the Human3.6M. Our Di²Pose achieves 49.2mm in average MPJPE, surpassing the performance of the SOTA diffusion model [22] by 0.5mm, which indicates that Di²Pose is able to enhance monocular 3D HPE in indoor scenes.

**3DPW.** Beyond indoor settings, we evaluate the performance of Di²Pose on the in-the-wild 3DPW dataset. As Table 2 shows, Di²Pose achieves the SOTA performance, and outperforms the SOTA method [22] by 3.4mm in MPJPE and 3.7mm in PA-MPJPE. On the occlusion-centric **3DPW-Occ**, Di²Pose maintains its superiority. When assessed under the 3DPW-AdvOcc protocol, all methods exhibit performance drops—MPJPE surges by up to 129% and PA-MPJPE by up to 72%. Despite this, Di²Pose remains markedly robust, leading the SOTA by significant margins in both MPJPE and PA-MPJPE, underscoring its effectiveness in handling occlusions.

**Qualitative Results.** Figure 4 presents the qualitative results of DiffPose [22] in comparison with our Di²Pose across two datasets. It can be observed that our method yields more accurate predictions than compared diffusion model (DiffPose), especially under various occlusion scenarios (self-occlusion and object occlusion). This demonstrates the superior occlusion-robustness of our Di²Pose.

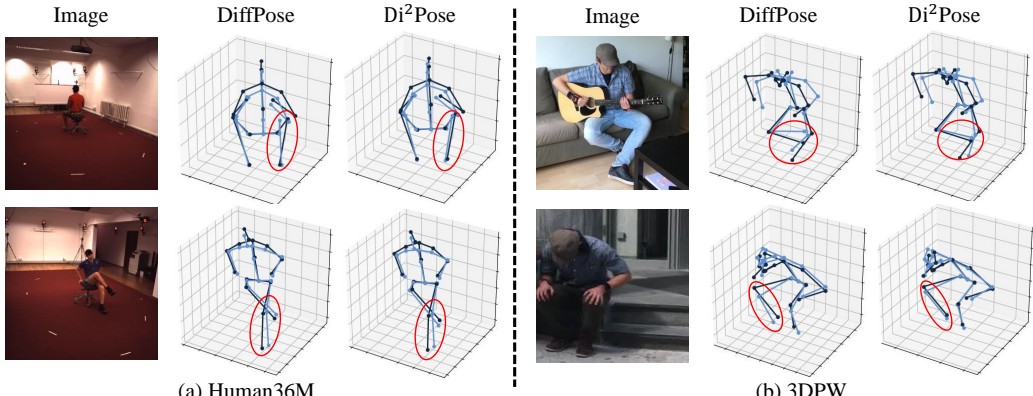

|  | Image | DiffPose | Di²Pose |  | Image | DiffPose | Di²Pose |
|---|---|---|---|---|---|---|---|

(a) Human36M

(b) 3DPW

Figure 4: Qualitative results on two datasets. The black lines represent the ground truth poses and the blue lines are prediction results.

Table 3: Ablations on Human3.6M. P-1 and P-2 represent MPJPE and PA-MPJPE, respectively.

| Loc. Num. | P-1 | P-2 | | Levels | P-1 | P-2 | | Occ. Rate | P-1 | P-2 | | | | Training Steps | |
|---|---|---|---|---|---|---|---|---|---|---|---|---|---|---|---|
| | | | | | | | | | | | | | | | |
| | | | | | | | | 0 | 50.4 | 39.3 | | | 25 | 50 | 100 |
| 1 | 14.5 | 13.0 | | $[8,5,5,5]$ | 15.2 | 15.9 | | 0.3 | 50.7 | 39.3 | Inference Steps | 25 | 51.7 | 51.1 | 50.3 |
| 3 | **13.6** | **12.5** | | $[7,5,5,5,5]$ | **13.6** | **12.5** | | 0.6 | 49.5 | 39.1 | | 50 | — | 50.6 | 49.9 |
| 5 | 14.1 | 12.8 | | $[8,8,8,6,5]$ | 13.8 | 12.7 | | 0.9 | **49.2** | **39.0** | | 100 | — | — | **49.2** |
| | | | | | | | | 1.0 | 51.0 | 39.5 | | | | | |

(a) Different local joint number $X$ of Joint Shift operations in JS-Block.

(b) Different levels per channel $[L_1, \cdots, L_d]$ of quantization process FSQ.

(c) Different final occlude rate $\overline{\gamma}_S$ for the occlude and replace transition matrix.

(d) Different number of training and inference steps $S$. P-1 are reported.

## 4.4 Ablation Study

**Effectiveness of Pose Quantization Step.** Our pose quantization step, which consists of Local-MLP blocks, is designed for representing 3D human pose by capturing the local interactions between 3D joints. Table 4 displays the MPJPE metrics comparing the original 3D poses with those reconstructed via various methods. The results show that our pose quantization step reconstructs 3D poses with lower errors compared to previous method [21], which uses an MLP-Mixer for global joint information extraction. It indicates

Table 4: Different representation methods for 3D HPE.

| Pose Repr. | MPJPE | PA-MPJPE |
|---|---|---|
| PCT [21] | 15.2 | 15.9 |
| Ours | **13.6** | **12.5** |

that our model learns a more accurate representation of 3D poses. In addition, we conducted other ablation studies to investigate different local joint numbers $X$ and levels per channel $[L_1, \cdots, L_d]$ within pose quantization step, as shown in Table 3a and Table 3b. As to different $X$, note that when $X = 1$, we only extract feature of individual joint, and when $X > 1$, JS-Block is able to capture local interactions of different joints. Experimental results indicate that $X = 3$ reaches lowest reconstruct error. As for $[L_1, \cdots, L_d]$, the best level of FSQ for pose quantization is $[7, 5, 5, 5, 5]$.

**Impact of Different Transition Matrices.** To demonstrate the effectiveness of the specifically designed occlude and replace transition matrix, we constructed three transition matrices for discrete diffusion process: occlude transition matrix, replace transition matrix, and occlude and replace transition matrix. Table 5 illustrates that the optimal results are achieved when utilizing the occlude and replace transition matrix. The suboptimal performance observed when exclusively employing the other two transition matrices can be attributed to the following reasons: Utilizing solely the replace transition matrix introduces the challenge of random, irrelevant sub-structures, complicating the learning of the reverse process; Conversely, relying exclusively on the occlude

Table 5: Different transition matrices for discrete diffusion model.

| Matrices | MPJPE | PA-MPJPE |
|---|---|---|
| Occlude | 51.0 | 39.5 |
| Replace | 50.4 | 39.3 |
| Both | **49.2** | **39.0** |

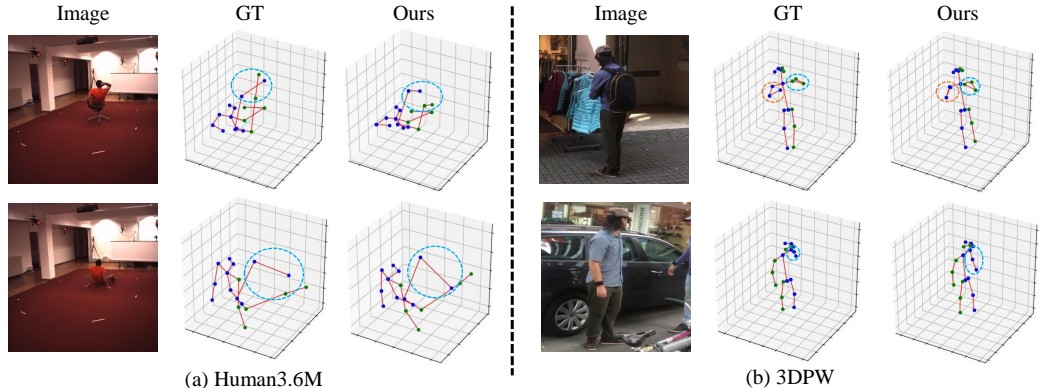

| Image | GT | Ours | Image | GT | Ours |

(a) Human3.6M                 (b) 3DPW

Figure 5: Failure cases of our Di$^2$Pose for 3D HPE. These instances primarily occur in scenarios with severe occlusions, as compared against ground truth (GT) poses. The content encircled by the dashed line indicates the parts where differences exist.

transition matrix causes the model to overly focus on the occluded portions, neglecting the contextual information from other visible parts. These clarifications can be verified in Table 3c, where we investigate the impact of different $\overline{\gamma}_S$. When $\overline{\gamma}_S = 0$, the occlude and replace transition matrix can be seen as the replace transition matrix, and when $\overline{\gamma}_S = 1$, the occlude and replace transition matrix can be seen as the occlude transition matrix. The best performance is obtained when $\overline{\gamma}_S = 0.9$.

In addition, we conducted an ablation study to investigate the impact of $S$ on the training and inference processes, as shown in Table 3d. We observed that using larger numbers of steps during both training and inference stages improves performance but also increases time complexity. Moreover, the results indicate that performance remains satisfactory even when the number of inference steps is reduced by 75% (e.g., from 100 steps during training to 25 steps during inference). This finding suggests a viable strategy for enhancing generation speed without significantly compromising quality.

## 5 Limitations

Figure 5 illustrates several results of 3D human pose estimation. When substantial occlusions cover the human body—obscuring the exact pose to the extent that it confounds even human observers—the predictions made by Di$^2$Pose may deviate from GT 3D pose. This deviation primarily stems from the inherent limitation of inferring 3D poses directly from 2D images, which lack critical spatial depth information. Such limitations introduce uncertainty and indeterminacy in the predictions.

Despite these challenges, Di$^2$Pose manages occlusions effectively by producing physically plausible outcomes. This capability is attributed to the integration of a pose quantization step within Di$^2$Pose, which constrains the model's search space to physically reasonable configurations. Note that the pose quantization step is trained on real 3D human pose data, enhancing its reliability under severe occlusions.

Currently, Di$^2$Pose is primarily designed for frame-based 3D HPE and does not utilize interframe data from videos. Future enhancements will focus on incorporating interframe information to refine the accuracy of 3D pose predictions further within the Di$^2$Pose framework.

## 6 Conclusion

This paper presents Di$^2$Pose, a novel diffusion-based framework that tackles occluded 3D HPE in discrete space. Di$^2$Pose first captures the local interactions of joints and represents a 3D pose by multiple quantized tokens. Then, the discrete diffusion process models the discrete tokens in latent space through a conditional diffusion model, which implicitly introduces occlusion into the modeling process for more reliable 3D HPE with occlusions. Experimental results show that our method surpasses the state-of-the-art approaches on three widely used benchmarks.

## Acknowledgments

This work was supported by the National Natural Science Foundation of China (62337001) and the Fundamental Research Funds for the Central Universities (226-2024-00058). Long Chen is supported by HKUST Special Support for Young Faculty (F0927) and HKUST Sports Science and Technology Research Grant (SSTRG24EG04).

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

## Appendix

In this Appendix, we provide relevant preliminary knowledge, mathematical proofs, complete training and inference algorithms, additional experimental results, more implementation details about our Di$^2$Pose and broader impacts.

## A Preliminary: Continuous Diffusion Model

The continuous diffusion model consists of two primary processes: the *forward process* and the *reverse process*. The forward process methodically corrupts the original data $x_0$ into a noisy latent variable $x_S$, which converges to a stationary distribution (e.g., a Gaussian distribution). Conversely, the reverse process aims to reconstruct the original data $x_0$ from $x_S$, utilizing learned parameters.

**Forward Process** Starting with $x_0$ drawn from the distribution $q(x_0)$, the forward process incrementally corrupts $x_0$ through a sequence of latent variables $x_{1:S} = (x_1, x_2, \ldots, x_S)$, where each $x_s$ retains the same dimensionality as $x_0$. This transformation is modeled as a fixed Markov chain:

$$q(x_{1:S}|x_0) = \prod_{s=1}^{S} q(x_s|x_{s-1}). \tag{12}$$

where each transition $q(x_s|x_{s-1})$ is defined by a Gaussian distribution:

$$q(x_s|x_{s-1}) = \mathcal{N}(x_s; \sqrt{1 - \eta_s}x_{s-1}, \eta_s I) \tag{13}$$

Here, $\eta_s$ is a small positive constant that follows a predefined schedule $(\eta_1, \eta_2, \ldots, \eta_S)$, allowing the data to progressively approach an isotropic Gaussian distribution, $\mathcal{N}(0, I)$, as $s$ increases. The overall transition from $x_0$ to $x_s$ can thus be expressed as:

$$q(x_s|x_0) = \mathcal{N}(x_s; \sqrt{\bar{\zeta}_s}x_0, (1 - \bar{\zeta}_s)I) \tag{14}$$

where $\zeta_s = 1 - \eta_s$ and $\bar{\zeta}_s = \prod_{i=1}^{s} \zeta_i$.

**Reverse Process** In the reverse process, the model aims to convert the latent variable $x_S$, which is assumed to follow the distribution $\mathcal{N}(0, I)$, back into the original data $x_0$. The joint probability distribution is given by:

$$p_\theta(x_{0:S}) = p(x_S) \prod_{s=1}^{S} p_\theta(x_{s-1}|x_s) \tag{15}$$

The conditional distributions involved are inferred using Bayes rule as follows:

$$q(x_{s-1}|x_s, x_0) = \frac{q(x_s|x_{s-1}, x_0)q(x_{s-1}|x_0)}{q(x_s|x_0)} \tag{16}$$

To optimize the generative model $p_\theta(x_0)$ for fitting the data distribution $q(x_0)$, we minimize a variational upper bound on the negative log-likelihood:

$$\mathcal{L}_{vb} = \mathbb{E}_{q(x_0)}\Big[D_{KL}[q(x_S|x_0)||p(x_S)] + \sum_{s=1}^{S} \mathbb{E}_{q(x_s|x_0)}\big[D_{KL}[q(x_{s-1}|x_s, x_0)||p_\theta(x_{s-1}|x_s)]\big]\Big]. \tag{17}$$

However, continuous diffusion models are not applicable in discrete spaces, such as quantized token indices $\mathbf{k} = (k_1, k_2, \ldots, k_N)$ where each $k_i$ assumes one of $|\mathcal{C}|$ discrete values. This limitation arises because Gaussian noise cannot corrupt discrete elements in a meaningful way. Thus, modeling in discrete spaces necessitates the development of discrete diffusion processes.

## B Mathematical Proofs

In this section, we provide a detailed mathematical proofs for Eq. (6), which can quickly calculate $q(\mathbf{k}_s|\mathbf{k}_0)$ according to Eq. (2).

Concretely, we use mathematical induction to prove Eq. (6). At first, we have following conditional information:

$$\alpha_s, \beta_s \in [0,1], \alpha_s = 1 - |\mathcal{C}|\beta_s - \gamma_s,$$

$$\overline{\alpha}_s = \prod_{i=1}^{s} \alpha_s, \overline{\gamma}_s = 1 - \prod_{i=1}^{s}(1-\gamma_s), \overline{\beta}_s = (1 - \overline{\alpha}_s - \overline{\gamma}_s)/|\mathcal{C}|. \tag{18}$$

Now we want to prove that $\overline{\mathbf{M}}_s c(\mathbf{k}_0) = \overline{\alpha}_s c(\mathbf{k}_0) + (\overline{\gamma}_s - \overline{\beta}_s)c(|\mathcal{C}| + 1) + \overline{\beta}_s$. Firstly, when $s = 1$, we have:

$$\overline{\mathbf{M}}_1 c(\mathbf{k}_0) = \begin{cases} \overline{\alpha}_1 + \overline{\beta}_1, & \mathbf{k} = \mathbf{k}_0 \\ \overline{\beta}_1, & \mathbf{k} \neq \mathbf{k}_0 \text{ and } \mathbf{k} \neq |\mathcal{C}| + 1 \\ \overline{\gamma}_1, & \mathbf{k} = |\mathcal{C}| + 1 \end{cases} \tag{19}$$

which is clearly hold. Suppose the Eq. (6) holds at step $s$, then for $s = s + 1$, we have:

$$\overline{\mathbf{M}}_{s+1} c(\mathbf{k}_0) = \mathbf{M}_{\mathbf{k}+1}\overline{\mathbf{M}}_t c(\mathbf{k}_0). \tag{20}$$

Now we consider three conditions:
(1) when $\mathbf{k} = \mathbf{k}_0$ in step $s + 1$, we have:

$$\begin{aligned}
\mathbf{M}_{s+1} c(\mathbf{k}_0)_{(\mathbf{k})} &= \overline{\beta}_s \beta_{s+1}(|\mathcal{C}| - 1) + (\alpha_{s+1} + \beta_{s+1})(\overline{\alpha}_s + \overline{\beta}_s) \\
&= \overline{\beta}_s(|\mathcal{C}|\beta_{s+1} + \alpha_{s+1}) + \overline{\alpha}_s(\alpha_{s+1} + \beta_{s+1}) \\
&= \frac{1}{|\mathcal{C}|}(\overline{\beta}_s(1 - \gamma_{s+1}) + \overline{\alpha}_s\beta_{s+1} - \overline{\beta}_{s+1}) * |\mathcal{C}| + \overline{\alpha}_{s+1} + \overline{\beta}_{s+1} \\
&= \frac{1}{|\mathcal{C}|}[(1 - \overline{\alpha}_s - \overline{\gamma}_s)(1 - \gamma_{s+1}) + |\mathcal{C}|\overline{\alpha}_s\beta_{s+1} - (1 - \overline{\alpha}_{s+1} - \overline{\gamma}_{s+1})] + \overline{\alpha}_{s+1} + \overline{\beta}_{s+1} \\
&= \frac{1}{|\mathcal{C}|}[(1 - \overline{\gamma}_{s+1}) - \overline{\alpha}_s(1 - \gamma_{s+1} - K\beta_{s+1}) - (1 - \overline{\gamma}_{s+1}) + \overline{\alpha}_{s+1}] + \overline{\alpha}_{s+1} + \overline{\beta}_{s+1} \\
&= \overline{\alpha}_{s+1} + \overline{\beta}_{s+1}.
\end{aligned} \tag{21}$$

(2) when $\mathbf{k} = |\mathcal{C}| + 1$ in step $s + 1$, we have:

$$\mathbf{M}_{s+1} c(\mathbf{k}_0)_{(\mathbf{k})} = \overline{\gamma}_s + (1 - \overline{\gamma}_s)\gamma_{s+1} = 1 - (1 - \overline{\gamma}_{s+1}) = \overline{\gamma}_{s+1}. \tag{22}$$

(3) when $\mathbf{k} \neq \mathbf{k}_0$ and $\mathbf{k} \neq |\mathcal{C}| + 1$ in step $s + 1$, we have:

$$\begin{aligned}
\mathbf{M}_{s+1} c(\mathbf{k}_0)_{(\mathbf{k})} &= \overline{\beta}_s(\alpha_{s+1} + \beta_{s+1}) + \overline{\beta}_s\beta_{s+1}(|\mathcal{C}| - 1) + \overline{\alpha}_s\beta_{s+1} \\
&= \overline{\beta}_s(\alpha_{s+1} + |\mathcal{C}|\beta_{s+1}) + \overline{\alpha}_s\beta_{s+1} \\
&= \frac{1 - \overline{\alpha}_s - \overline{\gamma}_s}{|\mathcal{C}|} * (1 - \gamma_{s+1}) + \overline{\alpha}_s\beta_{s+1} \\
&= \frac{1}{|\mathcal{C}|}(1 - \overline{\gamma}_{s+1}) + \overline{\alpha}_s(\beta_{s+1} - \frac{1 - \gamma_{s+1}}{|\mathcal{C}|}) \\
&= \overline{\beta}_{s+1}.
\end{aligned} \tag{23}$$

The proof of Eq. (6) is completed. Notably, according to Eq. (6), the computation cost of $q(\mathbf{k}_s|\mathbf{k}_0)$ can be reduced from $O(|\mathcal{C}|^2 S)$ to $O(|\mathcal{C}|)$.

## C    Algorithms for Discrete Diffusion Process

In this section, we provide complete training and inference algorithms for discrete diffusion process.

### C.1    Training Procedure

The discrete diffusion process aims to model quantized 3D pose tokens in a discrete space. This involves utilizing a 2D image $I$ and its corresponding 3D human pose $\mathbf{P}$ as inputs. The image $I$ serves as a contextual condition, while $\mathbf{P}$ is converted into discrete tokens for modeling.

---

**Algorithm 1** Training Algorithm for the discrete diffusion process.

---

**Require:**

A transition matrix $\mathbf{M}_s$, the number of steps $S$, parameters of pose denoiser $\theta$, training epoch $T$, pose dataset $\boldsymbol{D}$ (including 2D image $I$ and 3D human pose $\mathbf{P}$), and the well-learned pose encoder $f_{PE}(\cdot)$.

1: **for** $i = 1$ to $T$ **do**
2:     **for** $(I, \mathbf{P})$ in $\boldsymbol{D}$ **do**
3:         $\mathbf{k}_0 = \mathrm{FSQ}(f_{PE}(\mathbf{P}))$, $\boldsymbol{y} = \mathrm{ImageEncoder}(I)$;
4:         sample $s$ from $\mathrm{Uniform}\{1, 2, ..., S-1, S\}$;
5:         calculate $q(\mathbf{k}_s|\mathbf{k}_0)$ based on Eq. (6);
6:         estimate $f_\theta(\mathbf{k}_{s-1}|\mathbf{k}_s, \boldsymbol{y})$;
7:         calculate loss according to Eq. (10);
8:         update $\theta$;
9:     **end for**
10: **end for**
11: **return** $\theta$.

---

---

**Algorithm 2** Inference Algorithm for the discrete diffusion process.

---

**Require:**

The number of steps $S$, input 2D image $I$, the pose decoder $f_{PD}(\cdot)$, parameters of pose denoiser $\theta$, stationary distribution $p(\mathbf{k}_S)$;

1: $s = S$, $\boldsymbol{y} = \mathrm{ImageEncoder}(I)$;
2: sample $\mathbf{k}_s$ from $p(\mathbf{k}_S)$;
3: **while** $s > 0$ **do**
4:     $\mathbf{k}_s \leftarrow$ sample from $p_\theta(\mathbf{k}_{s-1}|\mathbf{k}_s, \boldsymbol{y})$
5:     $s \leftarrow (s-1)$
6: **end while**
7: **return** $f_{PD}(\mathbf{k}_s)$.

---

Firstly, the 3D human pose $\mathbf{P}$ is encoded by $f_{PE}(\cdot)$ and subsequently quantized using the FSQ technique, resulting in multiple discrete tokens. Concurrently, a pre-trained Image Encoder extracts contextual features from $I$, producing a conditional feature sequence $\boldsymbol{y}$. During the forward process, we sample $s$ from a uniform distribution $\{1, 2, ..., S-1, S\}$ and compute $q(\mathbf{k}_s|\mathbf{k}_0)$ based on Eq. (6). In the reverse process, the pose denoiser $f_\theta(\mathbf{k}_{s-1}|\mathbf{k}_s, \boldsymbol{y})$ is trained to estimate $q(\mathbf{k}_{s-1}|\mathbf{k}_s, \mathbf{k}_0)$. Finally, the overall loss is calculated according to Eq. (10), and the parameters of the pose denoiser $\theta$ are updated accordingly.

The complete training algorithm for the discrete diffusion process is presented in Algorithm 1.

### C.2 Inference Procedure

In the inference process, our objective is to recover the 3D human pose $\hat{\mathbf{P}}$ from an input 2D image and discrete tokens.

Initially, all pose tokens are either masked or initialized randomly, which is achieved by sampling from the stationary distribution $p(\mathbf{k}_S)$. The 2D image $I$ is encoded using the pre-trained Image Encoder. Subsequently, we predict $f_\theta(\mathbf{k}_{s-1}|\mathbf{k}_s, \boldsymbol{y})$ step by step until the pose tokens are fully recovered. Finally, the reconstructed tokens are decoded using the pose decoder $f_{PE}(\cdot)$, yielding the recovered 3D pose $\hat{\mathbf{P}}$.

The complete inference algorithm for the discrete diffusion process is presented in Algorithm 2.

## D Additional Implementation Details

All experiments are carried out on one NVIDIA A100 PCIe GPU. The proposed Di$^2$Pose is completely implemented in PyTorch [53]. In this section, we provide the detailed training settings for the pose quantization step and the discrete diffusion process.

Table 6: Results on Human3.6M in millimeters under PA-MPJPE. The best results are in bold, and the second-best ones are underlined.

| Methods | Dir | Disc | Eat | Greet | Phone | Photo | Pose | Pur | Sit | SitD | Smoke | Wait | WalkD | Walk | WalkT | Avg |
|---|---|---|---|---|---|---|---|---|---|---|---|---|---|---|---|---|
| Martinez et al. [48] ICCV'17 | 39.5 | 43.2 | 46.4 | 47.0 | 51.0 | 56.0 | 41.4 | 40.6 | 56.5 | 69.4 | 49.2 | 45.0 | 49.5 | 38.0 | 43.1 | 47.7 |
| Pavlakos et al. [54] CVPR'17 | 34.7 | 39.8 | 41.8 | **38.6** | 42.5 | 47.5 | 38.0 | 36.6 | 50.7 | 56.8 | 42.6 | 39.6 | 43.9 | 32.1 | 36.5 | 41.8 |
| Liu et al. [45] ECCV'18 | 35.9 | 40.0 | 38.0 | 41.5 | 42.5 | 51.4 | 37.8 | 36.0 | 48.6 | 56.6 | 41.8 | 38.3 | _42.7_ | 31.7 | 36.2 | 41.2 |
| Zhang et al. [82] TPAMI'23 | — | — | — | — | — | — | — | — | — | — | — | — | — | — | — | _39.1_ |
| Choi et al. [14] IROS'23 | 36.7 | 41.1 | 37.6 | 42.2 | 40.5 | 44.1 | 37.8 | 36.3 | 47.0 | 60.5 | 39.8 | 38.9 | 42.7 | 33.7 | 35.1 | 40.9 |
| Gong et al. [22] CVPR'23 | **33.9** | **38.2** | _36.0_ | _39.2_ | _40.2_ | **46.5** | _35.8_ | _34.8_ | 48.0 | _52.5_ | _41.2_ | _36.5_ | **40.9** | _30.3_ | _33.8_ | 39.2 |
| **Di²Pose (Ours)** | _34.5_ | _38.4_ | **35.1** | 40.8 | **39.8** | _47.0_ | **34.9** | **34.7** | **47.1** | **52.3** | **40.4** | **36.1** | 42.9 | **30.0** | **33.4** | **39.0** |

For the pose quantization step, we employ the AdamW [46] optimizer with $\beta_1 = 0.9$ and $\beta_2 = 0.999$, adhering to a base learning rate of 1e-3 and a weight decay parameter of 0.15. The training process is configured with a batch size of 256 across a total of 20 epochs.

For the discrete diffusion process, we still utilize the the AdamW optimizer with $\beta_1 = 0.9$ and $\beta_2 = 0.96$, adhering to a base learning rate of 5.5e-4 and a weight decay parameter of 4.5e-2. The training process is configured with a batch size of 64 across a total of 50 epochs.

# E    Additional Experimental Results

We exhibit more experimental results to verify the effectiveness of our Di²Pose.

## E.1    Quantitative Results

As shown in Table 6, we benchmark Di²Pose against SOTA 3D HPE methods on the Human3.6M under PA-MPJPE protocol. Our Di²Pose achieves 39.0mm in average PA-MPJPE, surpassing the performance of the compared SOTA 3D HPE methods, which indicates that Di²Pose is able to enhance monocular 3D HPE in indoor scenes.

## E.2    Qualitative Results

In this part, we present additional qualitative results on the Human3.6M and 3DPW datasets. As illustrated in Figure 6, our Di²Pose model demonstrates the ability to accurately recover 3D human poses in both indoor and in-the-wild scenarios. Particularly noteworthy is its performance under various occlusion conditions, including self-occlusion and object occlusion. Even in these challenging situations, Di²Pose consistently produces reasonable 3D pose estimations, highlighting its robustness to occlusions.

# F    Broader Impacts

This research focuses on estimating physically valid 3D human poses from monocular frames, especially under occlusion scenes. Such a method can be positively used for sports analysis, surveillance, healthcare, autonomous driving, etc. where clear, unobstructed views of the subject may not always be available. It can also lead to malicious use cases, such as illegal surveillance and video synthesis. Thus, it is essential to deploy these algorithms with care and make sure that the extracted human poses are with consent and not misused. Moreover, the diffusion-based model has a longer runtime compared to other CNN or GCN-based methods, causing more computational resources and energy consumption.

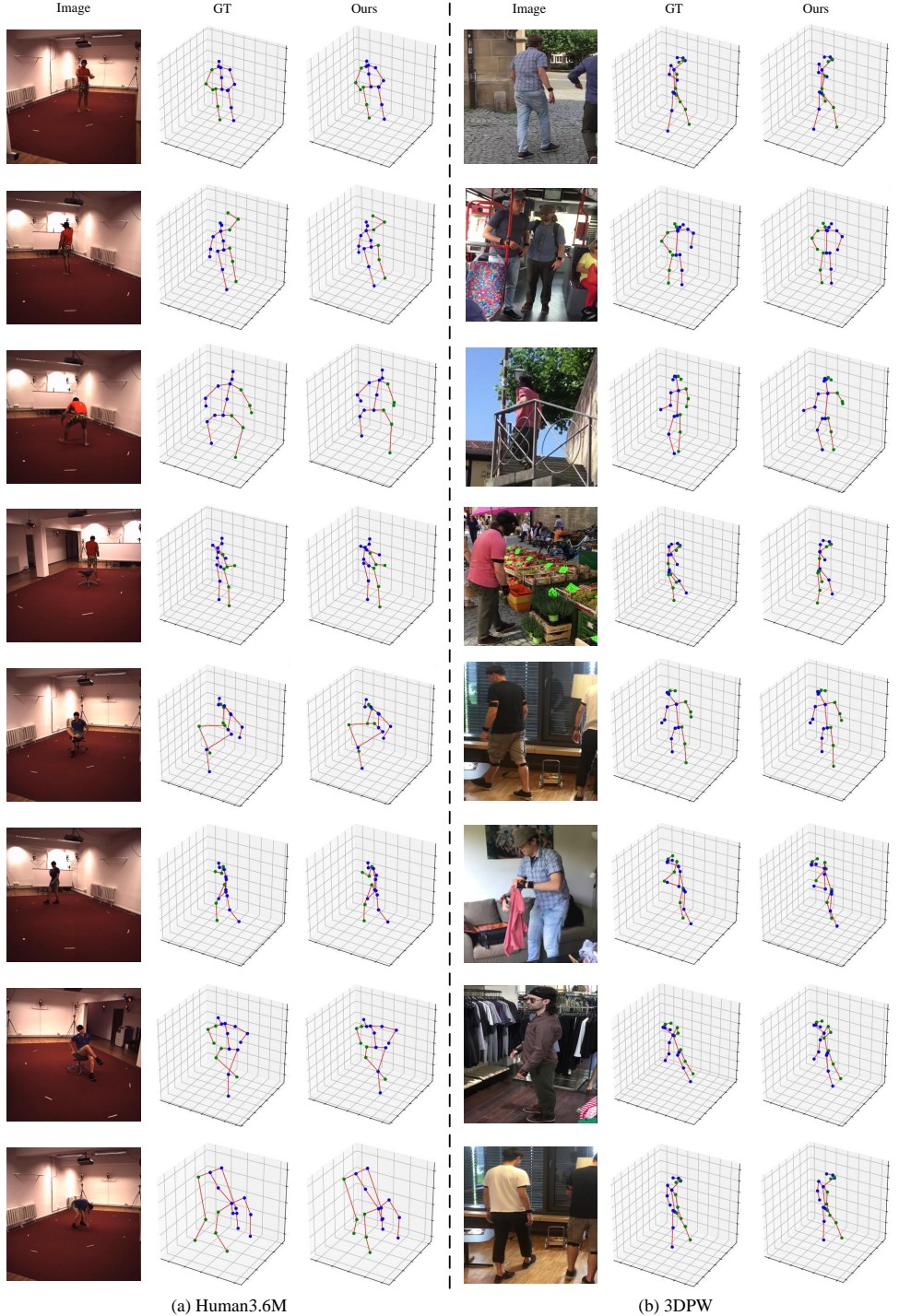

Figure 6: Qualitative results on two datasets. Joints on the right side are marked in green, while other joints are highlighted in blue.

