# OpenReview forum: "$\text{Di}^2\text{Pose}$: Discrete Diffusion Model for Occluded 3D Human Pose Estimation"
_NeurIPS.cc/2024/Conference — NeurIPS 2024 poster_

### Official Review · Reviewer_ACgE · 2024-06-21

**Soundness:** 2
**Presentation:** 3
**Contribution:** 2
**Rating:** 5
**Confidence:** 4

**Summary:**

This paper introduces a discrete diffusion method for 3D human pose estimation (HPE). Recent works have successfully applied diffusion models for HPE. However, they need a lot of training data and sometimes output non-anthropomorphic poses. In this work, the authors propose to use discrete diffusions, which leverage a quantized latent space and model the diffusion process as transitions between discrete states. The hope is that the result will be more constrained to the space of plausible poses as it can only take a limited number of values.

The main contributions of this paper are the following: (1) The authors propose a VQ-VAE for quantizing human pose and modeling the dependencies between the body joints; (2) A discrete diffusion model is designed to do HPE in the quantized latent space; (3) The "Occlude and replace" strategy is proposed to model occlusions, further improving the results.

**Strengths:**

- The approach is new: this is the first work using discrete diffusion for human pose estimation. This approach is promising and performs well, especially in the presence of occlusions.

- I found the paper quite well written. The weakness of prior works is clearly pointed out, and we understand how the authors hope to address them.

- There are plenty of details about the models and their mathematical ground. Calculations are well-detailed, which makes them easy to follow.

- The experiments and numerous ablations help capture the strengths of the introduced approach. The proposed approach outperforms SOTA approaches.

**Weaknesses:**

The method section could be improved to make it more clear:
- Giving the dimensions of $F$ and $T$ from the beginning (L137) would help understanding the VQ-VAE.
- (L183) I find it very surprising that the loss for training the VQ-VAE is a cross-entropy. Usually, we use reconstruction loss for training such models (here, it could have been an MSE or some MPJPE). It is unclear why this is not the case for the proposed model.
- (L185) It is unclear if $k$ is the sequence of $N$ tokens or only one token. Given the Equation 2, I guess it is the sequence?
- (L193) The notation $k_i$ is already used in Equation 2 (and I think this does not denote the same thing as here it is temporal).
- (L270) Using the same variable name for 2 different values is not a good practice.

Even if this paper is the first to apply discrete diffusion models to HPE, the methodology section does not bring much novelty. Section 3.2 could be considered preliminary as this directly applies [23]. From my understanding, the only difference is that the "Mask and replace" operation of [23] is renamed "Occlude and replace" (but it does the same thing). I think it is good to use prior works when designing a new method, but it should be made clear which part is new and which part is an application of prior works.

The introduction justifies this work by saying continuous diffusion models have high dimensions and that relationships between the joints are ignored. From my understanding, the latent representation of a pose is composed of $N=100$ tokens of dimension $d=5$. Each of the $d_i$ dimensions can take a certain number of values (respectively 7,5,5,5,5). That means that the codebook has 4375 tokens. The transition matrix $M$ between consecutive intermediate predictions is then of dimension $4375 \times 4375$ if we suppose that the $N$ tokens are independent (which is not supposed to be the case if we want the pose to be globally coherent). In the end, I am not sure that the proposed model's dimension is smaller than continuous diffusion models and that the relationships between the joints are better modeled.

I also have some doubts about the fairness of experiments. The proposed model uses a ViT backbone, while all others (except [20]) have CNN or GCN backbones with much fewer parameters. I believe this is a crucial component of the model and that it is impossible to evaluate the contributions if basic components differ from other approaches.

The material used for experiments is given, but this is valuable only if we have running time for experiments (which is not the case).

**Questions:**

- Is $k$ (L185) a sequence?
- Are there some changes between the proposed diffusion model and the model from [23]?
- Is the relationship between the $N=100$ tokens modeled?
- Is the plausibility of the results due to the quantization or the discrete diffusion process? I know this is linked, but for instance, would it be possible to do a latent diffusion process on the continuous latent representation of dimension $N \times d$ (which is much smaller) and do the quantization just before decoding?
- Were other backbones tested to quantify the role of this component in the results?
- What are the running times (training and testing)? Is it comparable to other methods?
- In Section 4, where are the error bars and the information about the statistical significance? Were the experiments run multiple times?
- One of the advantages of diffusion models is that they can generate diverse outputs given a single 2D observation, which helps address the ambiguity of predicting 3D from a single 2D POV and occlusions. However, no analysis is performed on that; it seems that a single prediction is made given an image. Is there a reason for that?

**Limitations:**

The section on the limitations is quite limited. I believe the problem of predictions under heavy occlusion could have been solved by making multiple predictions given an image, which is straightforward with generative models such as diffusions. Limitations could also mention the running time of the diffusion-based approaches. Limitations include the fact that the experiments ran for a single time, which is particularly concerning since diffusion models introduce stochasticity in predictions.

The societal impact could include environmental considerations.

---

> ### Author Rebuttal · Authors · 2024-08-07
>
> ## Q1: Clarifications for Method
>
> 1. The dimensions of **F** and **T** are provided from the beginning (L137).
> 2. We have re-examined our code and confirm that the **L1 loss function** ($L_{PQ}=||P-{\hat{P}}||_1$) was indeed used throughout our experiments. While this was a typographical error in the manuscript, it did not affect our results or conclusions.
> 3. **Different ‘k’ in the paper**.
>     - **$k_i$** (Eq.2, **italicized, not bold**) is a scalar, specifically the $i$-th entry to the codebook.
>     - **$\mathbf{k}$** (Eq.2 and L185, **non-italicized, bold**) is a vector, denoting the token indices which are quantized from token features by FSQ.
>     - **$\mathbf{k}_s$** (L193), where $s\in\{0,1,...,S\}$, is the token indices $\mathbf{k}$ at different discrete step $s$.
>
>     While the same letter ‘k’ is used, the distinctions are made clear by **the use of italics, bolding, and subscripts**. Moreover, the consistent use of $\mathbf{k}$ across two stages ensures uniformity and avoids using unnecessary new symbols.
>
> 4. **Model the relationship between 𝑁=100 tokens?** We only model the relationship between adjacent joints within a sub-structure, as shown in Fig._R 1 in pdf. However, we do not further model the relationship between different sub-structures.
>
> **We will update all these typos and further polish our presentation.**
>
> ## Q2: Differences between Di²Pose and [1]
>
> Admittedly, Di²Pose and [1] follow a mainstream scheme that utilizes a quantization step combined with a diffusion process, explored in various domains [2-4]. However, Di²Pose addresses a different problem with specific motivations and objectives.
>
> - **Distinct Motivation and Problem Domain:** Di²Pose aims to solve **occluded 3D HPE** by leveraging the discrete nature of 3D poses and the strengths of diffusion models in ***handling uncertainty and indeterminacy***. Our goal is to enhance the robustness and accuracy of 3D HPE under occlusion.
>
>     In contrast, [1] focuses on **text-to-image generation task**, specifically ***addressing the weaknesses of Autoregressive models*** like DALLE.
>
> - **Distinct Mechanism:** While the "Mask and replace" in [1] and our "Occlude and replace" share a similar transition matrix implementation, **their purposes are fundamentally different**. The strategy in [1] is designed to ***mitigate unidirectional bias and accumulated prediction errors***. Our pose quantization step produces a token sequence representing pose substructures, enabling the "Occlude and Replace" mechanism. **"Occlude" simulates the occlusion of a substructure**, and **"Replace" solves the uncertainty under occlusions**, where a single occluded region may correspond to multiple potential 3D poses. This effectively simulates the transition from occluded to recovered, integrating occlusion impacts into the estimation process.
> - **Contribution to the Field:** Di²Pose provides ***a new paradigm for tackling occluded 3D HPE***. By designing a specific pose quantization step and leveraging discrete diffusion tailored for this task, we contribute a novel approach for addressing occlusions effectively.
>
> We acknowledge the importance of distinguishing our contributions from prior work and ***will clarify and discuss these differences in the Method section***.
>
> ## Q3: Dimensions of search space
>
> In the introduction, we mention that continuous diffusion models need a larger search space to achieve optimal generative outcomes, whereas discrete diffusion models do not. It is noteworthy that our emphasis is on the size of the **search space** rather than the size of the **model's dimension**. For clarification, please **refer to the detailed analysis in “A general response to common concern” in the *global response***.
>
> ## Q4: Extra experiments
>
> - **Replace backbone**: We add experiments replacing the ViT backbone with a CNN-based backbone [6] and evaluate on the 3DPW. As shown in **Table_R 2** in pdf, it shows that Di²Pose maintains its superiority even with a CNN backbone.
> - **Repeated experiments**: We repeat experiments three times and evaluate on the 3DPW. As shown in **Table_R 3** in pdf, we report the “Mean $\pm$ Std” of MPJPE and PA-MPJPE across multiple runs. It shows robustness of Di²Pose, and the mean results still achieves SOTA.
> - **Multiple inference results**: In our paper, we focused on single inference outputs for practical 3D HPE applications. However, we acknowledge the diversity of outputs possible with diffusion models due to different initializations. Thus, we add experiments with multiple inferences from different initializations. The “Mean $\pm$ Std” is reported in **Table_R 3** in pdf, which shows that Di²Pose produces relatively stable results.
>
>     Additionally, we **visualize** the diverse outputs of Di²Pose from a single input image in **Fig._R 4** in pdf, demonstrating the model's ability to generate varied predictions under occlusion.
>
> - **Running times**: The training time and inference speed are shown in **Table_R 4** in pdf.
>
> ## Q5: Continuous latent representation + Quantization before decoding?
>
> The idea is indeed very interesting. However, this specific approach is beyond the scope of our current paper. We appreciate your comments and will consider exploring this idea in the future.
>
> ## Q6: Limitation
>
> **Environmental considerations**: We add potential negative environmental impacts as follow. The diffusion-based model has a longer runtime compared to other CNN or GCN-based methods, causing more computational resources and energy consumption.
>
> [1] Vector Quantized Diffusion Model for Text-to-Image Synthesis. CVPR 2022
>
> [2] Global Context with Discrete Diffusion in Vector Quantised Modelling for Image Generation. ICCV 2022
>
> [3] Priority-centric human motion generation in discrete latent space. ICCV 2023
>
> [4] Layoutdm: Discrete diffusion model for controllable layout generation. CVPR 2023
>
> [5] Deep High-Resolution Representation Learning for Human Pose Estimation. CVPR 2019

---

> > ### Comment · Reviewer_ACgE · 2024-08-10
> >
> > Thanks a lot to the authors for this rebuttal, which addressed many of my concerns. However, some of them are still unsolved:
> >
> > # Q1
> > 1) I cannot see the dimensions of $F$ and $T$ L137... The dimension of $F$ is given indirectly L149, and the dimension of T is unknown until L178.
> >
> > 3) Thank you for this explanation. I understand the different $k$ better now. I believe using different letters would be even clearer, but it is ok to keep it like that.
> >
> > 4) One of the motivations for the proposed method is to model the dependencies between the body joints (L32). Why is it ok to ignore the dependencies between sub-structures?
> >
> > # Q2
> > I understand that the application is completely different. However, the model is the same.\
> > I believe it is a great idea to use discrete diffusions for 3D HPE, and the components initially proposed for image generation do have a nice interpretation to solve occlusions.\
> > The main reproach I make is that the method section is presented in a way that lets the reader believe that all these techniques (for instance, the "Occlude and replace") mechanisms were never done before for discrete diffusions, which is not true.

---

> ### Author Response · Authors · 2024-08-11
> **Replying to Official Comment by Reviewer ACgE**
>
> We sincerely appreciate your thorough consideration of our responses and are delighted to address all questions and concerns.
>
> ## Q1:Dimensions of 𝐹 and 𝑇
>
> We apologize for this misunderstanding. The confusion arose due to the word limit in our rebuttal, which prevented us from clearly illustrating this point. We originally intended to convey that we fully agree with your suggestion, and we will revise the original manuscript to include explicit descriptions of the dimensions of $\mathbf{F}$ and $\mathbf{T}$ (e.g., ) starting from the beginning (L137), such as $\mathbf{F} = (\mathbf{f}_1, \mathbf{f}_2, \cdots, \mathbf{f}_N)$ ($\mathbf{f}_i \in \mathbb{R}^{D}$), and  $\mathbf{T}=(\mathbf{t}_1, \mathbf{t}_2, \cdots, \mathbf{t}_N)$ ($\mathbf{t}_i \in \mathbb{R}^{D}$) . This will make our presentation clearer and more intuitive.
>
> ## Q2: Dependencies between sub-structures?
>
> **We agree that considering the dependencies between sub-structures is an important problem**. Our approach follows the ***codebook learning paradigm (i.e., the encoder-decoder framework in VQ-VAE style)***, where an encoder encode the complete pose into discrete tokens, and a decoder decodes these tokens back to a reconstrcuted pose. ***Within this encoder-decoder framework, the interdependencies among sub-structures are mainly considered by the decoder, e.g., the discrete tokens (local sub-structure) can be reconstructed into the original 3D pose (global structure)***. Since our pose quantization step adheres to this pipeline, we have not implemented explicit strategies to further enhance the dependencies between sub-structures.
>
> **The motivation behind the design of the Local-MLP (L143-L145):** Our primary goal was to address a specific limitation observed in previous work [20 in the original paper], where MLPs were used to globally model the relationships between all joints. In the context of Di²Pose, our focus is on learning tokens that effectively represent different pose sub-structures. For each token, the key is to capture the relationships between the joints within a sub-structure. This is why we designed the Local-MLP to specifically model the dependencies between adjacent joints within each sub-structure.
>
> ## Q3: Reply to the comment on "Q2"
>
> We sincerely appreciate your insights on this matter. **We completely agree with your point**, and we have acknowledged this in our rebuttal where we stated, “We acknowledge the importance of distinguishing our contributions from prior work and will clarify and discuss these differences in the Method section.”
>
> Regarding your concern about the presentation of the method section, we would like to clarify that when writing this paper, we recognized that discrete diffusion models are a popular and widely-used framework. Our "Occlude and Replace" mechanism was specifically designed for occluded 3D HPE, and we chose this name to reflect its targeted application. However, we did not sufficiently discuss previous discrete diffusion model techniques in our original manuscript. This oversight could indeed cause some confusion, and we sincerely apologize for that.
>
> **To address the issues mentioned above, we will provide several revisions as follows:**
>
> 1. ***In the Related Work section***, we will include a discussion of the current development and applications of discrete diffusion models (such as [1-4]), highlighting both the connections and differences between our work and prior research.
> 2. ***In the Method section***, to avoid potential confusion, we will clearly state that **our "Occlude and Replace" transition matrix is inspired by prior work** [1]. We will also emphasize the distinctions and specific contributions of our approach in comparison.
>
> Once again, we are grateful for your constructive comments, which have greatly contributed to improving the clarity and quality of our paper. If you have any further questions or concerns, please don't hesitate to discuss them with us.
>
> [1] Vector Quantized Diffusion Model for Text-to-Image Synthesis. CVPR 2022
>
> [2] Global Context with Discrete Diffusion in Vector Quantised Modelling for Image Generation. ICCV 2022
>
> [3] Priority-centric human motion generation in discrete latent space. ICCV 2023
>
> [4] Layoutdm: Discrete diffusion model for controllable layout generation. CVPR 2023

---

> > ### Comment · Reviewer_ACgE · 2024-08-13
> >
> > Thanks a lot for this detailed answer. I increase my rating as most of my concerns are addressed.
> >
> > However, I am still not entirely convinced by the technical novelty. Is there any technical difference between the model of [23] and DI^2Pose? From my understanding of the paper and our discussion, I feel the only difference is the application domain.\
> > What is the difference between "Occlude and replace" and "Mask and replace"? I agree that this mechanism is well-suited to the problem of occluded 3D HPE, but changing its name does not make it a contribution if it works exactly like "Mask and replace" in [23].

---

> > > ### Author Response · Authors · 2024-08-14
> > > **Thanks for reviewing our work**
> > >
> > > We would like to express our sincere gratitude to you for your thorough review of our work. It is our pleasure to have addressed most of your concerns, and we greatly appreciate your recognition of our efforts by increasing the rating.
> > >
> > > We acknowledge that the "Occlude and Replace" mechanism shares similarities with the "Mask and Replace" technique from [23] on a technical level. However, our key contribution is being the first to effectively apply the discrete diffusion model to the occluded 3D HPE task, using the "Occlude and Replace" mechanism to simulate the full process of a 3D pose transitioning from occluded to recovered. Moreover, directly applying this technique in 3D HPE is not straightforward. The success of our approach depends on learning effective quantized tokens that represent sub-structures of 3D poses. Thus, another key contribution is the integration of the discrete diffusion process with a specifically designed pose quantization step, which together provide an effective solution for occluded 3D HPE.
> > >
> > > Once again, we sincerely thank you for your review and insightful suggestions, which have been instrumental in helping us improve the quality of our work.

---

### Official Review · Reviewer_s6UY · 2024-06-29

**Soundness:** 3
**Presentation:** 3
**Contribution:** 4
**Rating:** 6
**Confidence:** 3

**Summary:**

The work aims to solve the occlusion in 3D human pose estimation, which is an interesting and inherent topic in this field. The authors critiqued that the current continuous diffusion-based pose estimation method requires a large amount of data in the training, while 3D pose datasets are commonly insufficient and thus, hurt the accurate pose generation, especially for occluded cases. To address the problem, the authors first designed a Local-mlp block to quantize poses into tokens, while being aware of the local connection between each keypoints. Then, a diffusion model is exploited to learn the generation of tokens in the condition of a given image. To better address the occluded case, the author designed two addition tokens named "occ" to simulate the occlusion. All combined, the method shows on-par results in 3D pose estimations and the state-of-art performance in the occluded dataset.

**Strengths:**

1. The design of the "occ" tokens is interesting. Previous data augmentation is majorly in the image level, i.e. blocking a few key points in the image by adding noise or set to black. However, this paper does it at the token-level, which potentially leads to a more natural simulation of the occlusion.

2. Combining diffusion with 3D poses is an interesting topic. Especially generate the pose at a token-level, which leverages a similar idea to latent diffusion.

3. The paper is well-written and easy to follow.

**Weaknesses:**

1. In the introduction, the authors criticized the previous method, which requires a large amount of 3D pose data in training, while the proposed solution is to use the quantization of 3D poses. It is unclear why intuitively using a quantized representation of poses could be a solution to data dependency.

2. The target of this paper is also unclear. In my understanding of the author's argument, it is the data dependency of the discrete diffusion model that leads to inaccurate pose estimation in occlusion. However, instead of addressing the data dependency problem, the authors develop occ. token and replacement strategy to address the occlusion problem. This is confusing to me. It is like claiming there is a reason A for problem P, but instead of addressing A, the authors propose another method B completely irrelevant to A to address P from another perspective.

3. Confusing definitions of "Continuous" and "Discrete". There are two pairs of "Continuous" and "Discrete" in the paper. The first pair is the continuous and discrete representation of 3D poses. The second pair is a continuous and discrete diffusion model, as current DDPM or DDIM could be considered as a discretization of a continuous SDE or ODE. It is confusing why it is called continuous in the paper.

**Questions:**

1. For local-map, could there be a more clear explanation of how it exactly captures the interrelation between points?

2. Is it possible to visualize a few quantitative samples in the training process? Especially for poses that get replaced by occ tokens or some tokens get transferred to different tokens. I suspect such a replacement is potentially to be a good method to simulate the person-to-person occlusions.

3. What is the performance of directly using VQ-VAE to encode the 3D poses? Without these results, it is difficult to know the effectiveness of the proposed local-mlp.

**Limitations:**

No potential negative societal impact of the work is found.

---

> ### Author Rebuttal · Authors · 2024-08-07
>
> Thank you for the detailed comments. We are willing to address all your questions.
>
> ## Q1: Effectiveness of pose quantization for addressing data dependency
>
> The pose quantization step is designed to convert a 3D pose into multiple quantized tokens, which can be modeled in the latent space by the discrete diffusion model. The keypoint of addressing data dependency is that leveraging discrete diffusion process model discrete tokens rather than exploiting continuous diffusion model diffuses 3D pose in continuous space. For clarification, **we provide the detailed analyse in the global response**. Please refer to “**A general response to common concern”**.
>
> ## Q2: Clarification for the target of our paper
>
> First, we want to emphasize that **the primary goal** of our Di²Pose framework is to **address the occlusion problem in 3D HPE**. The framework consists of two stages, both designed to tackle occlusions effectively:
>
> - **Pose Quantization Step** transforms a 3D pose into multiple quantized tokens. Our Local-MLP and FSQ mechanisms are introduced to learn a better codebook, *ensuring that each quantized token represents an effective substructure of the pose*. This step is crucial for the subsequent discrete diffusion process, where the Occlude and Replace strategy is applied. By encoding poses into meaningful tokens, we facilitate the handling of occlusions using the discrete diffusion model.
> - During **Discrete Diffusion Process**, we use the Occlude and Replace strategy to address occlusions. Each Occ token represents an occluded sub-structure, while the replacement mechanism mitigates uncertainty. This process relies on the quantized tokens learned in Stage 1, ensuring that the model can simulate occlusions and accurately recover the complete pose.
>
> Regarding the data dependency issue, our discussion aims to highlight **an additional bonus** of using the discrete diffusion model over continuous ones. **This is beneficial but secondary to our main objective of solving the occlusion problem**.
>
> ## Q3: Clarification for the definitions of "Continuous" and "Discrete"
>
> **Discrete Representation of 3D Poses:** In this paper, we focus on the discrete representation of 3D poses. As described in the Introduction, existing 3D HPE methods tend to represent poses using ***coordinate vectors*** or ***heatmap embeddings***. These methods are both considered as discrete representations of 3D poses. Our ***pose quantization step*** converts a 3D pose into multiple discrete tokens, which still belong to discrete representations.
>
> **Continuous and Discrete Diffusion Models:** For the continuous and discrete diffusion models, we distinguish them by their initialization patterns at the beginning of the reverse process:
>
> - **Continuous Diffusion Model:** Continuous diffusion models initialize the 3D pose from random noise, where each joint can be sampled from the continuous 3D space. This aligns with the general understanding of continuous diffusion processes, such as those modeled by continuous SDEs or ODEs.
> - **Discrete Diffusion Model:** The discrete diffusion model, on the other hand, initializes the 3D pose from limited quantized tokens. The range of each token index depends on the size of the codebook, which is a finite number.
>
> ## Q4: Clear explanation of the Local-MLP
>
> We have redrawed Figure 3 from the original paper with more detailed structures, as shown in **Fig._R 1** in pdf.
>
> In Local-MLP, the key component is the JS-Block, which captures local interactions among X joints. Both Linear Proj. 1 and Linear Proj. 2 are implemented as Conv1d layers with a stride and padding of 1, which allows for information fusion along the joint dimension.
>
> 1. **Linear Proj. 1** integrates features within each joint in $\mathbf{P_{emb}^\top} \in \mathbb{R}^{D \times J}$.
> 2. **Joint Shift Operation** shifts features of adjacent joints into the same channel (Figure b(1) → Figure b(2)). The central part (green box) remains stationary, while the adjacent parts (blue and orange boxes) shift in opposite directions.
> 3. We extract the section indicated by the red dashed box (Figure b(2) → Figure b(3)), resulting in each green dashed box **containing features of different adjacent joints**.
> 4. **Linear Proj. 2** integrates features within each green dashed box, combining adjacent joint features (Figure b(3) → Figure b(4)).
> 5. **Channel MLP** further integrates information across each channel by expanding the dimension D by a factor of 4 and then mapping it back to D dimensions.
>
> ## Q5: Visualization of “Replace” mechanism
>
> We visualize the relevant results in **Fig._R 2** in pdf. First, Di²Pose uses the pose quantization step to represent a 3D pose as multiple discrete tokens, where each number represents an index of the codebook. To demonstrate the effect of the "Replace" operation in the discrete diffusion process, we ***replace a token at a specific position with other available tokens***, while ***keeping the tokens at other positions unchanged***. The resulting token sequence is then decoded by the Pose decoder to obtain a 3D pose. It can be seen that **replacing a certain token with other available tokens consistently changes the same sub-structure**, which is circled in red.
>
> ## Q6: Ablation studies on the Local-MLP
>
> We have addressed this issue in the **Ablation Study of our paper**.
>
> In **Table 3(a)**, we present the ablation study results for different local joint numbers X in the Joint Shift operations. It is noteworthy that the case where ***X=1 corresponds to not using the joint shift operation at all, which is regarded as a vanilla VQ-VAE***. The results indicate that our Local-MLP, which incorporates the joint shift operation, provides better pose representation performance.
>
> ## Q7: Potential negative societal impact
>
> In **Appendix F.2**, we have discussed the potential negative societal impacts of our work, including ***the risk of malicious applications such as illegal surveillance and video synthesis***.

---

> > ### Comment · Reviewer_s6UY · 2024-08-12
> >
> > Thank you for your detailed reply. The response has largely addressed my concerns. However, I have one additional concern, which I have outlined below.
> >
> > Regarding Q3, I disagree with the statement that “Continuous diffusion models initialize the 3D pose from random noise, where each joint can be sampled from the continuous 3D space. This aligns with the general understanding of continuous diffusion processes, such as those modeled by continuous SDEs or ODEs.”
> >
> > In my understanding, the term “continuous” in continuous diffusion models refers to the time variable t, not the space of the joints (either inputs or conditions). Take a traditional diffusion model DDPM as an example, which transitions pure white noise X_T at time T to X_0 at time 0, the “continuous diffusion” refers to considering X_t as a continuous function over time t, i.e., from X_t to X(t). The transition from X_{t} to X_{t-1} then becomes an SDE between two continuous functions X(t) and X(t-1) rather than a recursive formula. This concept is not about where X is sampled no matter this X is sampled from a continuous space (e.g., a multivariate Gaussian distribution) or a discrete space (e.g discrete latent space). I believe this difference in understanding of “continuous” may cause confusion among readers from the field of probabilistic generative models. I suggest that the authors provide further clarification on this point.
> >
> > Regarding the final rating, I noticed that the ratings are quite diverse. However, I place the most value on the review from reviewer ACgE, as another 7-level acceptance appears to have been autonomously generated by ChatGPT, which I am unsure whether to consider as a valid reference.
> >
> > Follow the discussion between the authors and reviewer ACgE, my major concern lies in the novelty of the work, as the “Occlude and Replacement” technique is not originally from this paper. Applying or adapting an existing method to a specific domain offers a different level of novelty compared to proposing an entirely new method. Therefore, although I am still inclined to support acceptance, I would like to adjust my rating to “weakly accept.” I would also be interested in hearing the responses from reviewer tz5g and any further response from ACgE on the issue of novelty.

---

> > > ### Author Response · Authors · 2024-08-14
> > > **Thanks for reviewing our work**
> > >
> > > Thank you for your continued support and for providing a thoughtful assessment of our work. We appreciate your inclination to support acceptance, and we value your feedback on the relevant statements regarding “continuous diffusion.”
> > >
> > > ### **Avoid confusion about “continuous diffusion”**
> > >
> > > We fully agree with your suggestion that the term "continuous" as used in the context of “continuous diffusion” may cause confusion to readers. To avoid misunderstanding, we will revise our statements in Introduction as follows:
> > >
> > > > Prior diffusion-based 3D HPE methods initialize the 3D pose from random noise at the begining of the diffusion process, where each joint can be sampled from the continuous 3D space. Since the continuous 3D space has an infinite number of points, training such diffusion-based models requires a large amount of 3D pose data to achieve optimal outcomes.
> > > >
> > >
> > > This clarification should help readers directly grasp the point we intend to convey and avoid any confusion regarding the “continuous diffusion”. In addition, we will revise other relevant parts about “continuous diffusion” in our paper accordingly to avoid confusing.
> > >
> > > ### **Clarify our contributions**
> > >
> > > Regarding the novelty of our work, we acknowledge that the "Occlude and Replace" mechanism shares similarities with prior techniques. However, our key contribution is being the first to effectively apply the discrete diffusion model to the occluded 3D HPE task, using the "Occlude and Replace" mechanism to simulate the full process of a 3D pose transitioning from occluded to recovered. Moreover, directly applying this technique in 3D HPE is not straightforward. The success of our approach depends on learning effective quantized tokens that represent sub-structures of 3D poses. Thus, another key contribution is the integration of the discrete diffusion process with a specifically designed pose quantization step, which together provide an effective solution for occluded 3D HPE.
> > >
> > > Once again, we appreciate your thoughtful feedback and suggestions, which have been instrumental in refining our work.

---

### Official Review · Reviewer_tz5G · 2024-07-09

**Soundness:** 2
**Presentation:** 4
**Contribution:** 2
**Rating:** 5
**Confidence:** 4

**Summary:**

This work claims that the 3D human pose of a single frame is discrete, and learns the local pairing relationship between joint points to generate the human pose under occlusion. At the method level, VQ-VAE is used for human skeleton quantization, and then combined with the diffusion model to solve this discrete relationship. The main contributions are as follows: 1. The Di2Pose framework is proposed, which integrates the inherent discreteness of 3D pose data into the diffusion model, providing a new paradigm for 3D HPE under occlusion. 2. The designed pose quantization step represents the 3D pose in a combinatorial manner, effectively captures the local correlation between joints, and limits the search space to reasonable configurations. 3. The constructed discrete diffusion process simulates the complete process of 3D pose from occlusion to recovery, and introduces the impact of occlusion into the pose estimation process.

**Strengths:**

1. The structure and writing of the article are relatively clear.
2. The inherent discreteness of 3D pose data is integrated into the diffusion model. Although this is a downstream task of VQ-VAE+Diffusion, it is indeed a new idea for 3D HPE of monocular 2D images.
3. The reasoning process of Di^2Pose in Chapter 3 is very clear, which helps to understand the principle of the overall method.
3. The influence of occlusion is introduced into the pose estimation process, and the codebook that encodes the human pose is back-diffused, which is also a solution to occluded human pose.

**Weaknesses:**

1. From an intuitive level, the human skeleton in this work still uses the simplest H3.6M skeleton with only 17 joints, not even the version with 32 key points. The error tolerance rate is very high for pose estimation of models with such a small number of joints. There is not even any obvious joint change. The Average MPJPE value in the Human3.6M list of 17 joint skeletons has stabilized within 30. The human skeleton in this work needs to use more complex SMPL, SMPL-X, and STAR to see the effect better.
2. The image and skeleton prediction are independent. I hope the author can render the skeleton into the image and map it in the image to better observe the subtle differences in movement. It is difficult to observe the 3D skeleton alone, especially in Figure 4. I can't even find any obvious difference in the human posture in Figure a. In addition, when estimating a single-frame 3D skeleton, it is best to give results from multiple perspectives. I think the perspectives shown in the article are too casual.
3. Although the method is clearly written, I think Di^2Pose is just a downstream task of VQ-VAE+Diffusion. In terms of design, it is not even as clever as [1]. I think Di^2Pose is just DiffPose with the addition of VQ-VAE.
4. The dataset used in the experiment is too outdated. I hope to see more new datasets for verification. Like EMDB[2], SLOPER4D[3], CIMI4D[4], etc.

[1] Feng H, Ma W, Gao Q, et al. Stratified Avatar Generation from Sparse Observations[C]//Proceedings of the IEEE/CVF Conference on Computer Vision and Pattern Recognition. 2024: 153-163.
[2] Kaufmann M, Song J, Guo C, et al. Emdb: The electromagnetic database of global 3d human pose and shape in the wild[C]//Proceedings of the IEEE/CVF International Conference on Computer Vision. 2023: 14632-14643.
[3] Yan M, Wang X, Dai Y, et al. Cimi4d: A large multimodal climbing motion dataset under human-scene interactions[C]//Proceedings of the IEEE/CVF Conference on Computer Vision and Pattern Recognition. 2023: 12977-12988.
[4] Dai Y, Lin Y T, Lin X P, et al. Sloper4d: A scene-aware dataset for global 4d human pose estimation in urban environments[C]//Proceedings of the IEEE/CVF conference on computer vision and pattern recognition. 2023: 682-692.

**Questions:**

I hope to see the results on new datasets and SMPL renders to images. I will consider improving my score.

**Limitations:**

See weaknesses.

---

> ### Author Rebuttal · Authors · 2024-08-07
>
> Thank you for the detailed comments. We are willing to address all your questions.
>
> ## Q1: Extended experiments on more complex datasets
>
> **Clarification**. In 3D skeleton-based HPE task, the **17-joint annotation** of the Human3.6M is a **widely-used and standard benchmark**. Most mainstream methods validate their methods on this dataset. For **fair comparisons**, we also follwed this setting in our experiments. As described in [3], this limited number of joints helps discard the smallest links associated to details for the hands and feet, going as far down the kinematic chain to only reach the wrist and the ankle joints." This has led subsequent works to predominantly use the 17 joints for experiments rather than the 32 joints.
>
> **Extra results on a new dataset**: We acknowledge the reviewer's concern that the 17-joint Human3.6M dataset is relatively simple. Thus, we utilize a recent **H3WB dataset** [4], an extended annotation of Human3.6M with 133 keypoints covering the body, hands, and face, significantly increasing complexity. The experimental results are shown in **Table_R 1** in pdf. Di²Pose still achieves comparable results with SOTAs in terms of “All” and “Body”, but slightly underperforms on the "Face" and "Hand". This may be because the joints within the “face" and "Hand" are densely distributed and highly correlated, unlike the torso. Separate pose quantization for "Face," "Hand," and "Body" might be needed. Due to the limited rebuttal time, we did not have the opportunity to try such idea.
>
> **Di²Pose with SMPL, SMPL-X, and STAR** and **add experiments on new mesh-based datasets?** It is important to clarify the differences between skeleton-based (3D coordinates) and mesh-based (e.g., SMPL) 3D HPE tasks. Di²Pose is specifically designed for skeleton-based HPE. SMPL models, on the other hand, require estimating both pose and shape parameters, necessitating a redesign of existing Di²Pose. This level of complexity is beyond the scope of the current paper. However, we are very interested in adapting Di²Pose for mesh-based models. We will try to explore this direction in future work.
>
> ## Q2: Better visualizations
>
> **Render the 3D skeleton into the image?** In 3D skeleton-based HPE, projecting the 3D skeleton onto a 2D plane can **lead to overlapping joints and visual confusion**, making it difficult to assess estimation quality. This is different from mesh-based methods, where shape parameters allow for more accurate visual alignment with human silhouettes. Consequently, skeleton-based methods usually visualize results directly in 3D space [1,2].
>
> **Comparison with GT in the same 3D space**. To solve the reviewer's concern about the difficulty in discerning differences in human poses. We visualize the predictions and ground truth in the same 3D space, as shown in **Fig._R 5** in pdf.
>
> **Visualization on multiple views**: We add visualization results on multiple viewpoints, as shown in **Fig._R 3** in pdf.
>
> **We will add these visualizations in the final version.**
>
> ## Q3: Clarification for the question about “VQ-VAE+Diffusion”
>
> Although Di²Pose follows the “codebook learning + discrete diffusion process” pipeline, it is not as simple as “VQ-VAE + DiffPose”.
>
> - **Codebook learning:** The primary challenge in the first stage is to *learn effective and representative tokens, with each representing a sub-structure*. Simply applying vanilla VQ-VAE does not capture the relationships between adjacent joints within a sub-structure and is prone to causing codebook collapse. These issues would render the subsequent discrete diffusion process ineffective due to invalid tokens, preventing occlusion simulation and recovery. Specifically, we design the Local-MLP and exploit FSQ to ensure successful sub-structure representations.
> - **Discrete Diffusion Process:** While our second stage indeed involves a diffusion process, it differs significantly from *DiffPose, which employs a continuous diffusion model*. Di²Pose operates on a **discrete token sequence**, while DiffPose diffuses **3D poses in continuous space**. Moreover, Our process uses a **transition matrix** for state transitions, while DiffPose employs a **Gaussian Mixture Model** to model the uncertainty distribution.
>
> Importantly, **Di²Pose seamlessly integrates both stages to address occluded 3D HPE**. Quantized tokens from the first stage are processed by the discrete diffusion model to simulate occlusions and recover the full pose. This mechanism effectively incorporates occlusions into the HPE process, offering valuable insights for tackling the occluded 3D HPE task.
>
> **Comparsion with [5]:** Regarding the comment "In terms of design, it is not even as clever as [1]," we would like to highlight the following points:
>
> - Thanks for bring this paper to our attention. Firstly, we want to mention that this paper [5] was released to arXiv on May 30, 2024, which is after the NeurIPS submission deadline (May 22, 2024).
> - Both Di²Pose and [5] utilize a pipeline involving codebook learning and a diffusion process. However, **[5] focuses on body-motion generation**, a task involving sequential signals that ***require inter-frame information***. In contrast, Di²Pose addresses occluded 3D HPE, targeting ***single-frame setting***. The differences in the problems being solved naturally lead to different design choices and frameworks, making a direct comparison of the methods' cleverness inappropriate. We will add these discussions and comparisons to our paper.
>
> [1] Diffusion-Based 3D Human Pose Estimation with Multi-Hypothesis Aggregation. In ICCV, 2023
>
> [2] GLA-GCN: Global-local Adaptive Graph Convolutional Network for 3D Human Pose Estimation from Monocular Video. In ICCV, 2023
>
> [3] Human3. 6m: Large scale datasets and predictive methods for 3d human sensing in natural environments. In TPAMI, 2013
>
> [4] H3WB: Human3.6M 3D WholeBody Dataset and Benchmark. In ICCV. 2023
>
> [5] Stratified Avatar Generation from Sparse Observations. In CVPR. 2024

---

> > ### Comment · Reviewer_tz5G · 2024-08-13
> >
> > Thanks for the author's reply.
> >
> > ---
> > - Regarding the question of skeleton-based and mesh-based methods, what I want to express is that skeletons can be redirected to achieve migration between skeletons. Just like the H3.6M skeleton with 17 joints can be migrated to the SMPL skeleton with 24 key points. Many papers have implemented such methods, which is no longer a difficult problem.
> >
> > - The reason why I hope to see a mesh-based human body model is that the skeleton method is monotonous in visualization. Predicting 10 more Beta parameters can take the visualization effect to a higher level, which I think is necessary to implement.
> >
> > - Discrete expression is the focus of this paper, and the network structure also determines the tone of discrete data. Although the author has made a comprehensive explanation, just like the target detection task in autonomous driving, the single-frame-based method will never be able to obtain more and more coherent information than the continuous-frame-based method. This is also my concern.
> >
> > ---
> > This paper is excellent in terms of writing and methodological integrity, but like the old school papers, I would like to see more reasonable and interesting innovations.

---

> > > ### Author Response · Authors · 2024-08-14
> > > **Thanks for reviewing our work**
> > >
> > > Thanks for your thoughtful consideration and feedback.
> > >
> > > We fully acknowledge the point you raised about the possibility of redirecting skeletons to achieve migration between different skeleton models. We also agree that extending our Di²Pose framework to mesh-based 3D HPE is a feasible direction. However, due to the structural differences between skeleton and mesh models, this still require certain modifications to our framework, such as redesigning the pose quantization step to incorporate shape parameters and other aspects specific to SMPL or similar models. This extension would require structural redesign and extensive hyperparameter tuning. Due to the limited time of rebuttal period, we do not have enough time to fully explore these aspects. However, we acknowledge its importance and consider it a valuable direction for future work.
> > >
> > > Regarding the concern about the single-frame-based method not capturing as much coherent information as continuous-frame-based methods, we agree that continuous-frame approaches inherently have an advantage in this regard. However, our single-frame-based framework can serve as a foundational step for future work that extends to continuous-frame-based methods, which would incorporate temporal coherence and continuity into more complex and comprehensive frameworks.
> > >
> > > Once again, we sincerely appreciate your thorough considerations, which have been instrumental in refining our work.

---

### Official Review · Reviewer_Z93t · 2024-07-16

**Soundness:** 3
**Presentation:** 3
**Contribution:** 2
**Rating:** 7
**Confidence:** 3

**Summary:**

The paper presents  novel diffusion-based framework for occluded 3D Human Pose Estimation (HPE) that operates in discrete space. Di2Pose leverages a two-stage process: a pose quantization step and a discrete diffusion process. The pose quantization step captures the local interactions between joints and represents the 3D pose as multiple quantized tokens. These tokens are then modeled in the latent space through a discrete diffusion process. This approach allows the framework to effectively manage occlusions by simulating the transition of a 3D pose from occluded to recovered, enhancing the reliability of pose estimation under occlusion conditions. Extensive evaluations on benchmarks like Human3.6M, 3DPW, and 3DPW-Occ demonstrate that Di2Pose outperforms state-of-the-art methods, particularly in occluded scenarios.

**Strengths:**

Introduction of Di2Pose Framework
   Di2Pose integrates the inherent discreteness of 3D pose data into the diffusion model, providing a novel paradigm for addressing 3D HPE under occlusions. This framework leverages a two-stage process involving pose quantization and discrete diffusion to confine the search space to physically plausible configurations and simulate the transition from occluded to recovered poses.

Pose Quantization Step:
   The designed pose quantization step effectively captures local correlations between joints by representing 3D poses in a compositional manner. This step confines the search space to reasonable configurations by learning from real 3D human poses, ensuring that the model generates physically plausible poses even under severe occlusions.

Discrete Diffusion Process
   The discrete diffusion process simulates the complete transition of a 3D pose from occluded to recovered, incorporating the impact of occlusions into the pose estimation process. This process models the quantized pose tokens in latent space, enhancing the model’s capability to understand and predict occluded parts of the human pose.

Strengths claimed and shown include

Effectiveness in Occluded Scenarios: Di2Pose demonstrates significant improvements in 3D HPE accuracy under occlusions compared to state-of-the-art methods, highlighting its superior occlusion-handling capabilities.

Physically Plausible Pose Generation: By integrating pose quantization and discrete diffusion, Di2Pose confines the search space to physically reasonable configurations, ensuring the generation of biomechanically valid poses even in challenging occluded scenarios.

Comprehensive Evaluation: The framework has been extensively evaluated on multiple challenging benchmarks, consistently yielding lower errors and demonstrating its robustness and generalizability across different datasets.

Use of Discrete Diffusion: The introduction of a discrete diffusion process tailored for 3D HPE provides a new perspective in the field, aligning more closely with the inherent discreteness of 3D pose data.

These contributions collectively position Di2Pose as a robust and innovative solution for 3D HPE, particularly in the presence of occlusions, advancing the state-of-the-art in this challenging domain.

**Weaknesses:**

Mechanistic Insights:
   The paper does not delve deeply into the mechanistic aspects of how the discrete diffusion process works in conjunction with pose quantization. Detailed explanations of the underlying mechanisms and how they contribute to the observed improvements would enhance understanding.

Comparative Benchmarking:
   While Di2Pose shows improvements over existing methods, a direct comparison with other contemporary methods such as the Pose Relation Transformer (Chi et al., ICRA 2023) and InfoGCN (Chi et al., CVPR 2022) would provide a clearer picture of its relative strengths and weaknesses.

Citation of Related Works:
   Incorporating references to other successful approaches in the field, such as the above

 These citations would position Di2Pose within the broader context of ongoing research and highlight its unique contributions relative to other leading methods.

Addressing these areas would strengthen the paper, providing deeper insights into the framework’s operation and situating it more firmly within the current landscape of 3D HPE research.

**Questions:**

See above

**Limitations:**

Limitations are not explored or discussed. It follows the same formula - compare against a benchmark and do ablation studies and state it is better than state of art.

---

> ### Author Rebuttal · Authors · 2024-08-07
>
> Thank you for the detailed comments. We are willing to address all your questions.
>
> ## Q1: Detailed explanations and insights about the two main parts: pose quantization and discrete diffusion.
>
> The proposed Di²Pose is a two-stage framework designed to address the challenges of occluded 3D human pose estimation (HPE).
>
> - **In the first stage**, we train a pose quantization step that transforms a 3D pose  $\mathbf{P}$ into $N$ discrete tokens $\mathbf{k}$. Each token represents a sub-structure of the whole pose. This pose quantization step leverages the discrete nature of 3D poses and **represents them as quantized tokens** by capturing the local interactions between joints. This quantization step is crucial for the subsequent discrete diffusion process, as *it allows the discrete diffusion model to simulate occlusions of specific sub-structures of the 3D pose*. The quantized tokens $\mathbf{k}$ **serve as a vital link that binds pose quantization and the discrete diffusion process together**, ensuring coherent interaction between the two stages.
> - **In the second stage**, we model tokens in the discrete space using a discrete diffusion process, which consists of a forward and a reverse process.
>
>     **Forward Process:** During the forward process, each token of $\mathbf{k}$ is probabilistically occluded with an Occ. token or replaced with another available token. The **occluded token** represents the *occlusion of the corresponding sub-structure of the 3D pose*. The **token replacement** mechanism is designed to *enhance the diversity of potential sub-structures*, reflecting the indeterminacy in occluded parts.
>
>     **Reverse Process:** In the reverse process, the pose tokens are initially occluded or randomly initialized. The denoising diffusion process estimates the probability density of pose tokens step-by-step based on the input 2D image until the tokens are completely reconstructed. **Each step leverages contextual information from all tokens of the whole pose as predicted in the previous step**, facilitating the estimation of a new probability density distribution and the prediction of the current step’s tokens. This sequential approach ensures a detailed and accurate reconstruction of 3D poses from occluded scenes.
>
>
> This two-stage framework allows Di²Pose to effectively handle occlusions by breaking down the pose into meaningful sub-structures and reconstructing them through a probabilistic diffusion process. The integration of pose quantization with the discrete diffusion process significantly contributes to the observed improvements in handling occluded poses.
>
> ## Q2: Comparison with related works
>
> We compare our Di²Pose with the mentioned related works as follows.
>
> - **Pose Relation Transformer (PORT) [1]:** PORT is designed to address the HPE problem, mitigating the effect of occlusions inspired by the sentence completion task in NLP. The **similarity** between PORT and our Di²Pose is that *both methods recognize the importance of capturing local context between adjacent joints*. PORT leverages an attention mechanism to aggregate adjacent joint features, while Di²Pose proposes a Local-MLP to capture local relationships within a sub-structure of the 3D pose. Moreover, PORT introduces a Masked Joint Modeling (MJM) approach to reconstruct randomly masked joints, which helps refine occlusions.
>
>     However, MJM randomly selects joint indices for masking and trains PORT to reconstruct the masked joints, **explicitly simulating occlusions** and treating the masked joints as independent. In contrast, Di²Pose uses a discrete diffusion process to **implicitly model occlusion** within the latent space, enhancing its understanding of how occlusions affect human poses. Additionally, we found that **PORT is specifically designed for 2D HPE**, which **differs from our focus on 3D HPE**. Thus, we cannot directly compare these two methods by experiments.
>
> - **InfoGCN [2]:** InfoGCN is proposed to solve the human skeleton-based action recognition task. This novel method focuses on embedding physical constraints and intention information into the latent representations of human actions. The **similarity** between InfoGCN and Di²Pose is that *both methods aim to learn informative latent representations from raw data*. InfoGCN introduces a novel learning objective based on the information bottleneck theory, which aims to learn an efficiently compressed latent representation of an action. However, Di²Pose proposes a pose quantization step, which leverages VQ-VAE to convert a 3D pose into multiple discrete latent tokens, **compressing information in different ways**. Moreover, **InfoGCN focuses on the human skeleton-based action recognition task**, which **is different from 3D HPE**. Therefore, we also cannot directly compare our method with InforGCN by experiments.
>
> Although the above works focus on different tasks, we recognize the importance of discussing these methods to provide a comprehensive context for our work. To address this, **we will cite these two works in the Related Work section** and add these discussions  about their contributions and differences with our approach. Additionally, we will further investigate other relevant works to enhance the completeness and context of our study.
>
> ## Q3: Limitations
>
> We have illustrated the limitations of our method in detail in the **Appendix Sec.F**, including failure cases, physically reasonable outcomes, and frame-based limitations.
>
> [1] Pose Relation Transformer Refine Occlusions for Human Pose Estimation. In ICRA, 2023
>
> [2] InfoGCN: Representation Learning for Human Skeleton-based Action Recognition. In CVPR, 2022

---

### Author Rebuttal · Authors · 2024-08-07

We thank all reviewers for recognizing our paper well-written (Reviewers tz5G, s6UY, ACgE), easy to follow (Reviewers s6UY, ACgE),  and with novel ideas/methods (all Reviewers).

We appreciate their careful reviews and constructive comments. We have revised our paper according to all comments. The major changes are summarized as follows.

- **According to Reviewer Z93t’s comments**:
    - Detailed explanations and insights about our method. We detailedly illustrate the designs and insights of two main parts: pose quantization and discrete diffusion.
    - Comparison with related works. We compare our Di²Pose with two related works mentioned by the reviewer.  We will cite these works in the Related Work section and further enhance the completeness of our study.
- **According to Reviewer tz5G’s comments**:
    - Extended experiments. We clarify the differences between skeleton-based 3D HPE (our purpose) and mesh-based 3D HPE. Moreover, we add experiments on a new dataset (H3WB) with more keypoints to increase complexity (cf **Table_R 1** in **pdf**).
    - Better visualizations. We visualize the predictions and ground truth in the same 3D space (cf **Fig._R 5** in **pdf**) and we also add visualization results from multiple viewpoints (cf **Fig._R 3** in **pdf**).
    - Clear clarification. We illustrate the specific design of our two-stage model for occluded 3D HPE task and compare our Di²Pose to the simple “VQ-VAE + DiffPose”. In addition, we make acomparisons with the mentioned paper for clear distinction.
- **According to Reviewer s6UY’s comments**:
    - Clear clarifications. We make clearer clarifications to address the following concerns.
        - Effectiveness of pose quantization for addressing data dependency. We illustrate this by comparing the search space between continuous and discrete diffusion models.
        - Target of our paper. We emphasize that the primary goal of our Di²Pose framework is to address the occlusion problem in 3D HPE. An additional bonus of using the discrete diffusion model is its ability to alleviate data dependency issues.
        - Clarification for the definitions of "Continuous" and "Discrete". We provide detailed explanations of both definitions mentioned in our paper for better understanding.
    - Further explanation of the Local-MLP. We have redrawed Figure 3 from the original paper with more detailed structures of Local-MLP (cf **Fig._R 1** in **pdf**).
    - Visualizations. We add new visualizations about the “Replace” mechanism in the discrete diffusion process (cf **Fig._R 2** in **pdf**).
- **According to Reviewer ACgE’s comments**:
    - Clear clarifications. We provide clearer clarifications to address the following concerns.
        - Method section. We correct relevant typos and further polish our presentation.
        - Differences between Di²Pose and the mentioned method. We distinguish both methods from the perspectives of motivations, problem domains, and mechanisms. We also highlight the contribution of Di²Pose to the occluded 3D HPE field.
        - Search space comparisons. We emphasize that the strength of Di²Pose lies in the size of the search space rather than the model's dimensionality. We further compare the search space between continuous and discrete diffusion models.
    - Experiments. We add various experiments to enhance the completeness of our study.
        - Replace backbone: We add experiments replacing the ViT backbone with a CNN-based backbone (cf **Table_R 2** in **pdf**).
        - Repeated experiments: We repeat experiments for the entire training and inference process (cf **Table_R 3** in **pdf**).
        - Multiple inference results: We add experiments with multiple inferences from different initializations (cf **Table_R 3** in **pdf**). In addition, we visualize the diverse outputs of Di²Pose from a single input image (cf **Fig._R 4** in **pdf**).
        - Running times: We provid the training time and inference speed of our model (cf **Table_R 4** in **pdf**).
    - Limitation. We add the potential negative environmental impacts in the Limitation section.

## A general response to common concerns raised by Reviewer s6UY and Reviewer ACgE

We appreciate the detailed feedback from both reviewers. In the Introduction section, we illustrated that continuous diffusion models need a large search space to achieve optimal generative outcomes. This statement raised concerns from both reviewers regarding **whether the pose quantization of Di²Pose effectively addresses data dependency** and **whether Di²Pose has a smaller search space**. To address both questions, we would like to compare the search spaces of continuous diffusion models and the discrete diffusion model as follows:

**Search Space Comparison:** To understand this, let's consider the reverse process of the diffusion model. For continuous diffusion models, the initialization of the 3D pose is sampled randomly from the continuous 3D space, which means the theoretical search space is **{continuous 3D space}^{joint number}**. Since ***the continuous 3D space has an infinite number of points***, training such a continuous diffusion model requires a large amount of 3D pose data to achieve optimal outcomes.

In contrast, for the discrete diffusion model, we initialize a limited number of quantized tokens. For each token, the number of initialization choices is {codebook size+1}, where 1 represents the Occ token. Thus, the theoretical search space for the discrete diffusion model is **{codebook size+1}^{quantized token numbers}**. This finite search space significantly reduces the amount of 3D pose data required for training.

We hope the above analysis will address the reviewers' concerns.

---

### Decision · Program_Chairs · 2024-09-25

**Decision:**

Accept (poster)

**Comment:**

VQ-VAE followed by Diffusion has been applied to various tasks, but its use in 3D Human Pose Estimation is novel. As noted by Reviewers tz5G and z93t, this application introduces some ideas around using the Reverse discrete diffusion process to enhance robustness against occlusion and to ensure contextual consistency.

Reviewer tz5G raised a concern that the H3.6M dataset is not sufficiently challenging for demonstrating improved results on complex poses. Although the authors provided results on an extended annotated version of H3.6M in their rebuttal, this does not fully address the underlying simplicity of the data. Additionally, Reviewer z93t rightly points out the lack of mechanistic insights in the pose estimation process.

Nonetheless, the authors have given detailed responses in their rebuttals, leading some reviewers, such as ACgE, to upgrade their scores after considering the authors' clarifications, due to which I recommend an accept.